# Raindrop Size Distribution (DSD) during the Passage of a Tropical Cyclone NIVAR: Effect of Measuring Principle and Wind on DSDs and Retrieved Rain Integral and Polarimetric Parameters from Impact and Laser Disdrometers

Basivi Radhakrishna

National Atmospheric Research Laboratory, Department of Space, Govt. of India, Gadanki - 517112, Andhra Pradesh, India.

**Correspondence:** Basivi Radhakrishna (rakibasivi@gmail.com)

**Abstract.** Raindrop size distribution (DSD) observations during the passage of landfalling tropical cyclone NIVAR by impact (JWD) and laser (LPM and PARSIVEL) disdrometers are used to unveil the DSD characteristics in the eyewall, inner, and outer rainbands. Disdrometer measurements collected at the same location are used to study the effect of wind, measuring principle, and hardware processing on the DSDs and, in turn, on estimated rain integral and polarimetric parameters. The concentration of raindrops of diameters between 0.7 mm to 1.5 mm increases with rain rate (R) in all the regions of NIVAR, while the magnitude of the increase is high in the eyewall than in the inner and outer rainbands. The DSD characteristics reveal that for a given R, relatively large reflectivity (Z) and mass-weighted mean diameter ($D_m$) are found in the outer rainband and small Z, and $D_m$ in the eyewall than in other regions of a TC. Raindrops of diameter 3-mm in size are observed frequently in inner and outer rainbands, while infrequent in the eyewall at R greater than 5 mm h$^{-1}$. The DSDs and estimated rain integral and polarimetric parameters are distinctly different for various disdrometers at similar environmental conditions. Raindrops greater than 3 mm in size are infrequent in the JWD recordings while frequent in the LPM an PARSIVEL indicating LPM and PARSIVEL overestimates the raindrop size when the fall path deviates from nadir due to horizontal wind. The wind effect on the recorded DSD and estimated rain integral and polarimetric parameters are not uniform in various regions of NIVAR for different disdrometers as the measuring principle and hardware processing further influence these effects. Along with the differences in measured DSD spectra, the resonance effects at X-band for raindrops greater than 3-mm cause variations in the estimated polarimetric parameters between the disdrometers.

## 1 Introduction

Tropical cyclones (TCs) are destructive atmospheric phenomena associated with extremely high winds and ample rainfall, which cause severe damage to human life and the economy. The advancements made in recent years noticeably improved the numerical weather prediction (NWP) models that forecast of TCs genesis and tracks (Hendricks et al., 2011); however, intensity predictions are still to be improved (DeMaria et al., 2014). All scales (micro-scale to synoptic) of forcings influence the intensity fluctuations of a TC (Molinari and Vollaro, 1989; Bosart et al., 2000; Hanley et al., 2001); however, small-scale,

transient, moist convective processes and resultant latent heating play a major role in different regions (McFarquhar et al., 2006). Convective processes and resulting rainfall in a TC are primarily governed by the evolution of the microphysics of a TC (Khain et al., 2016). The microphysical process information is obtained by studying the raindrop size distribution (DSD). DSD is the raindrop concentration per drop size per unit volume. Spatiotemporal variations of DSD at various scales in different rain types are essential for disclosing the fundamental precipitation microphysical processes, including collision–coalescence, breakup, and evaporation (Rosenfeld and Ulbrich, 2003; Radhakrishna et al., 2020). Considering the vast application of DSD, it is one of the prime measurements required in the fields of remote sensing and numerical weather prediction. The differences in dynamical and microphysical processes from eyewall to inner rainbands to outer rainbands (Houze, 2010) cause changes in the DSD observed at the surface (Merceret, 1974; Homeyer et al., 2021). This shows the importance of DSD in various regions of a TC to better represent the microphysics in NWP models for improving the intensity predictions (Fierro and Mansell, 2017; Wang et al., 2020).

DSD varies in different regions of a TC (Merceret, 1974; Homeyer et al., 2021), seasonally, and from noncyclonic rain (Radhakrishna and Rao, 2010). Mass-weighted mean diameter ($D_m$) comparisons over Pacific (Chen et al., 2012), Atlantic (Tokay et al., 2008), and Bay of Bengal (Radhakrishna and Rao, 2010) basins show the largest $D_m$ values over the Bay of Bengal and smallest $D_m$ values over the Pacific than other basins. The studies mentioned above used different disdrometers (impact, video, and laser-based) to measure the DSD at the surface. The laser-based particle size velocity (PARSIVEL) disdrometer underestimates small raindrops (Tokay et al., 2014; Thurai et al., 2017; Wen et al., 2018) compared to a two-dimensional video disdrometer (2DVD). These differences in DSDs are due to variations in measuring principles of drop diameter by various disdrometers. The Joss-Waldvogel disdrometer (JWD) measures the drop size by measuring the impact of falling raindrops on a pressure sensor converted into an electric signal (Joss and Waldvogel, 1967). Laser precipitation monitor (LPM) and PARSIVEL disdrometers measure drop size by accounting for the variations in the intensity of laser beam between emitter and receiver (Illingworth and Stevens, 1987; Löffler-Mang and Joss, 2000). Two orthogonal line scan camera images of 2DVD provide raindrop size, shape, and velocity (Kruger and Krajewski, 2002). Each principle and hardware processing have its advantages and disadvantages, leading to errors and uncertainties in the measured DSD spectrum. 2DVD is considered the most reliable in measuring DSDs accurately (Raupach and Berne, 2015; Thurai et al., 2017); however, further works by Thurai and Bringi (2018), and Raupach et al. (2019) showed that these disdrometers underestimate small raindrops considerably.

The disdrometer evaluation experiment (DEVEX) showed a good agreement between PARSIVEL, 2DVD, and a dual-beam spectropluviometer (Krajewski et al., 2006). However, PARSIVEL measured more number of smaller drops and higher rainfall rates than the other two. Considering DSDs from TCs and organized mesoscale convective systems, Thurai et al. (2011) showed that PARSIVEL and 2DVD show good agreement till 20 mm h$^{-1}$, while PARSIVEL overestimates 20%-30% at higher rainfall rates. Krajewski et al. (2006) attributed these differences to instruments' background noise, condensation of water vapor on the lenses, splashes, and margin fallers. Tokay et al. (2014) compared JWD and PARSIVEL and showed good agreement in the DSD spectra above 0.5 mm diameter. Angulo-Martínez et al. (2018) and Guyot et al. (2019) found the recording of more number of smaller drops by LPM than PARSIVEL, and these errors are amplified with increasing rain intensity. Errors in DSD measurements are affected by instrument principle and associated hardware and external environmental conditions like wind

speed and direction (Friedrich et al., 2013; Capozzi et al., 2021). Strong wind conditions create turbulence along the walls of 2DVD, deflecting the small drop path, resulting in more intersects leading to an excess of smaller drops (Nešpor et al., 2000). To study the wind speed and direction effects on laser disdrometer, Friedrich et al. (2013) used articulating and stationary disdrometers and found marginal variations for small drops (< 2 mm). However, the articulating disdrometer recorded higher concentrations of large (> 5 mm; 200–500 m$^{-3}$ mm$^{-1}$) and medium-sized (2 - 5 mm; 500–3000 m$^{-3}$ mm$^{-1}$) drops compared to the stationary disdrometer.

Disdrometers are used as ground truth to validate the radar geophysical parameters. The artefacts and instrument errors associated with various kinds of disdrometers mentioned above need to be quantified as they propagate to the retrievals of radar geophysical parameters (Adirosi et al., 2018) and, in turn, in surface rainfall from weather radars (both polarimetric and non-polarimetric). Mitigating these errors is crucial for representing the microphysics in the NWP models correctly. Thus, considering all these artifacts and errors, the present study is aimed to study the differences in DSDs observed by JWD, PARSIVEL, and LPM in different regions of a landfalling very severe cyclonic storm NIVAR originated over the Bay of Bengal. Also, this study assesses the effect of horizontal wind speed on DSDs observed by impact and laser disdrometers and the retrieved rain integral and polarimetric parameters.

## 2 Disdrometers data processing

### 2.1. Joss-Waldvogel disdrometer

JWD is an impact-type disdrometer that measures raindrops primarily into 127 diameter classes, hitting a surface area of 50 cm$^2$ with an accuracy of 95% in a 1-minute time interval (Joss and Waldvogel, 1967). These 127 classes are further combined into 20 intervals, distributed more or less exponentially, measuring raindrops from 0.3 mm to 5.3 mm. JWD calssifies the DSD in each diameter interval is estimated from the 1-minute JWD observations as follows:

$$N(D_i) = \frac{10^6 * n_i}{F * t * v(D_i) * \Delta D_i} \ (m^{-3} \ mm^{-1}) \tag{1}$$

Where $i$ stands for the diameter interval number, $N(D_i)$ is the number of drops per unit volume per unit diameter interval, $F$ is the measuring area (5000 mm$^2$), $t$ is the sampling time (60 s), $n_i$ is the number of drops in the $i^{th}$ class interval, $D_i$ is the $i^{th}$ class equivolume diameter (mm), $v(D_i)$ is the fall velocity of the drop with diameter $D_i$ (m s$^{-1}$), and $\Delta D_i$ is the $i^{th}$ class drop interval (mm).

### 2.2. Thesis Clima laser precipitation monitor

Thesis Clima LPM uses a 228 mm length, 20 mm width, and 0.75 mm thickness laser bean of wavelength 780 nm with a resulting sampling area of 45.6 cm$^2$ (Illingworth and Stevens, 1987). However, the manufacturer will provide the information of slight variations in the dimensions of the laser beam for each disdrometer separately using a parameter called AU$_{parameter}$. Hence, the measuring area is device-specific for LPM and is estimated using the following equation.

$$F_{LPM} = \frac{4600 * 1000}{AU_{parameter}} \ (mm^2) \tag{2}$$

For the LPM used in this study, the $AU_{parameter}$ is 916 resulting in a sampling area of 50.218 cm$^2$. LPM measures raindrops between 0.18 mm and 8 mm in 22 different diameter intervals with 20 fall velocity intervals ranging from 0.1 m s$^{-1}$ to 10.5 m s$^{-1}$. LPM records data at a 1-minute resolution. The drops falling on the edges of the laser beam effectively reduce the sampling area of the LPM depending on the diameter of the drop (Löffler-Mang and Joss, 2000). The effective sampling area is given by

$$F_{LPM}^i = F_{LPM} * \frac{20 - D_i}{20} \ (mm^2) \tag{3}$$

The drop size distribution is estimated as follows:

$$N(D_i) = \frac{10^6}{t} * \sum_{j=1}^{20} \frac{n_{i,j}}{v_L(j) * D_i * F_{LPM}^i} \ (m^{-3} \ mm^{-1}) \tag{4}$$

Where $n_{i,j}$ is the number of drops recorded by LPM in $i^{th}$ diameter and $j^{th}$ velocity interval, $v_L(j)$ is the fall velocity of raindrop with diameter $D_i$ (m s$^{-1}$) measured by the LPM.

**2.3. OTT PARSIVEL disdrometer**

The second-generation PARSIVEL disdrometer manufactured by OTT Hydromet Inc consists of a 780 nm laser beam with dimensions of 180 mm length, 30 mm width, and 1 mm thickness providing a sampling area of 54 cm$^2$ (Löffler-Mang and Joss, 2000). PARSIVEL records raindrops in the range of 0.1 mm and 24.5 mm in 32 diameter and 32 velocity intervals (ranges between 0.05 and 20.8 m s$^{-1}$). PARSIVEL is also showing margin fallers; the effective sampling area and N(D) are calculated using the following relations.

$$F_{PARSIVEL}^i = 180 * (30 - 0.5 * D_i) \ (mm^2) \tag{5}$$

$$N(D_i) = \frac{10^6}{t} * \sum_{j=1}^{32} \frac{n_{i,j}}{v_P(j) * D_i * F_{PARSIVEL}^i} \ (m^{-3} \ mm^{-1}) \tag{6}$$

Where $v_P(j)$ is the fall velocity of raindrop with diameter $D_i$ (m s$^{-1}$) measured by the PARSIVEL. The processing done by the manufacturer of disdrometer converts the electrical signals into number of drops in each drop diameter interval. After obtaining the number of drops information in each diameter interval, equations (1) to (6) are used to estimated N(D) from the respective disdrometer.

## 2.4. Rain integral and polarimetric parameters

In open fields, JWD and LPM are installed 10 m apart (13.4608°N, 79.1733°E) while PARSIVEL is 500 m away from both in the southeast direction (13.4565°N, 79.1758°E). During the passage of NIVAR, JWD and PARSIVEL observations are available throughout the event, while LPM observations are available after 1415 IST on 25[th] November 2020. The disdrometer data are quality checked before estimating the rain integral and polarimetric parameters. The 1-minute data recordings are considered only when they show drop measurements in more than five diameter class intervals and the number of drops measured is greater than 50. This threshold condition removes the spurious values from the disdrometer recordings caused by non-precipitating targets (Radhakrishna and Rao, 2010). The splashing and margin filler effects are removed using velocity thresholds used in Jaffrain and Berne (2011), and Friedrich et al. (2013) for the laser disdrometers. The quality-controlled data are used to estimate the N(D) using (1), (4), and (6). The estimated N(D) is used to calculate rain rate (R), reflectivity (Z) assuming Rayleigh approximation, $D_m$, and normalized intercept parameter ($N_w$) using the following relations.

$$R = 3.6 * 10^{-3} * \frac{\pi}{6} * \sum_i [N(D_i) * D_i^3 * v(D_i) * \Delta D_i] \ (mm \ h^{-1}) \tag{7}$$

$$Z = \sum_i [N(D_i) * D_i^6 * \Delta D_i] \ (mm^6 \ m^{-3}) \tag{8}$$

$$D_m = \frac{\sum_i [N(D_i) * D_i^4 * \Delta D_i]}{\sum_i [N(D_i) * D_i^3 * \Delta D_i]} \ (mm) \tag{9}$$

$$N_w = \frac{4^4 * \sum_i [N(D_i) * D_i^3 * \Delta D_i]}{6 * D_m^4} \ (m^{-3} \ mm^{-1}) \tag{10}$$

The polarimetric parameters are estimated using scattering amplitudes from the T-matrix simulations (Mishchenko et al., 1996) at S- (2.8 GHz), C- (5.6 GHz), and X-band (9.3369 GHz, the frequency of X-band radar operating at Gadanki) frequencies. The scattering simulations are performed in the temperature ranges from 5 °C to 30 °C using the refractive index of raindrops estimated from Ray (1972) and the drop axis ratio relation from Brandes et al. (2002). The polarimetric radar parameters reflectivity in horizontal ($Z_H$) and vertical ($Z_V$) polarizations, differential reflectivity ($Z_{DR}$), specific differential phase ($K_{DP}$), the co-polar correlation coefficient between horizontal and vertical polarizations ($\rho_{HV}$), two-way specific differential attenuation ($A_{DP}$) and specific attenuation at horizontal polarization ($A_H$) are estimated using back-scattering ($S_{HH}$, $S_{VV}$) and forward scattering ($F_{HH}$, $F_{VV}$) amplitudes (in mm).

$$Z_{H,V} = \frac{4 * \lambda^4}{\pi^4 * |K^2|} * \sum_i [N(D_i) * |S_{HH,VV}^i|^2 * \Delta D_i] \ (mm^6 \ m^{-3}) \tag{11}$$

$$Z_{DR} = 10 * log_{10}\left(\frac{Z_H}{Z_V}\right) (dB) \tag{12}$$

$$K_{DP} = \frac{180 * \lambda * 10^{-3}}{\pi} * \sum_i [N(D_i) * Re(F_{HH}^i - F_{VV}^i) * \Delta D_i] \; (^\circ \; km^{-1}) \tag{13}$$

$$A_{DP} = 8.686 * \lambda * 10^{-3} * \sum_i [N(D_i) * Im(F_{HH}^i - F_{VV}^i) * \Delta D_i] \; (dB \; km^{-1}) \tag{14}$$

$$A_{H,V} = 8.686 * \lambda * 10^{-3} * \sum_i [N(D_i) * Im(F_{HH,VV}^i) * \Delta D_i] \; (dB \; km^{-1}) \tag{15}$$

$$\rho_{HV} = \left| \frac{\sum_i [N(D_i) * S_{VV}^i S_{HH}^{ic} * \Delta D_i]}{\{\sum_i [N(D_i) * S_{HH}^i S_{HH}^{ic} * \Delta D_i]\}^{1/2} * \{\sum_i [N(D_i) * S_{VV}^i S_{VV}^{ic} * \Delta D_i]\}^{1/2}} \right| \tag{16}$$

Where i stands for diameter interval, c on superscript indicates the complex conjugate, $\lambda$ is the wavelength (mm) considered, and K is the complex refractive index whose real part denotes the phase speed, and imaginary part indicates the extinction. $K_{DP}$ is immune to attenuation and is widely used to correct the attenuation using $K_{DP} - A_H$ and $K_{DP} - A_{DP}$ relations which is the advantage of polarimetric when compared to conventional weather radars (Bringi et al., 1990; Jameson, 1991; Park et al., 2005). The studies by Bringi et al. (1990) and Jameson (1991) showed that $K_{DP}$, $A_H$, and $A_{DP}$ are related linearly, while Park et al. (2005) showed a power-law relation. As the powers are $\sim 1$ over the Gadanki region (Rao et al., 2018), linear relations of $K_{DP}$, $A_H$, and $A_{DP}$ are considered following Bringi et al. (1990) and are given below.

$$A_{DP} = \gamma_{DP} * K_{DP} \tag{17}$$

$$A_H = \gamma_H * K_{DP} \tag{18}$$

$\gamma_{DP}$ is the differential attenuation coefficient, $\gamma_H$ is the attenuation coefficient, and both depend on the DSD characteristics, temperature, and drop shape. As $A_{DP}$ and $A_H$ are in $dB \; km^{-1}$, both $\gamma_{DP}$ and $\gamma_H$ are expressed in dB per degree.

## 3   DSD measurements during the NIVAR cyclone

On $21^{st}$ November 2020, a low-pressure area is formed over the equatorial Indian Ocean and adjoining central parts of the South Bay of Bengal. It concentrated into a depression over southwest and adjoining southeast Bay of Bengal on 0230 IST

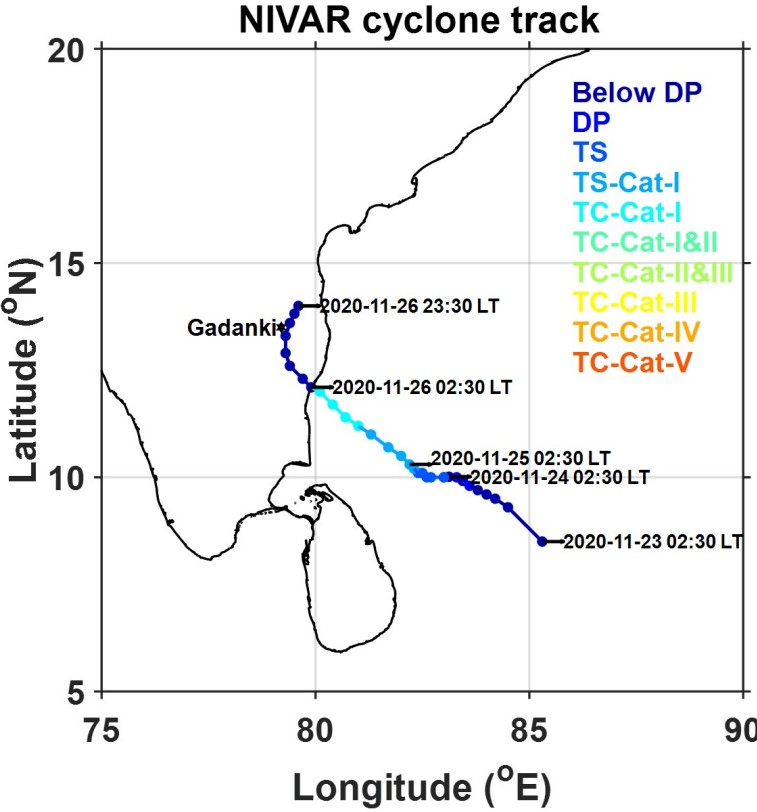

**Figure 1.** Track and category of NIVAR cyclone formed over the Bay of Bengal is shown with a time interval of 3 h. The cyclone categories are based on the Dvorak classification. The black star indicates the location of Gadanki where disdrometers are installed.

on $23^{rd}$ and moved west-northwestwards, and intensified into a deep depression in the evening of the same day. It is further

intensified into a cyclonic storm 'NIVAR' over southwest Bay of Bengal at 0530 IST on $24^{th}$. It moved in the same direction and intensified into a severe cyclonic storm at midnight (2330 IST) on the $24^{th}$ and into a very severe cyclonic storm in the afternoon (1430 IST) on the $25^{th}$. Moving further northwestwards, NIVAR made landfall at 2330 IST on $25^{th}$ at 12.1°N and 79.9°E near Puducherry as a very severe cyclonic storm with a wind speed of 120 kmph. After landfall, it moved further northwestwards and weakened into a severe cyclonic storm at 0230 IST on $26^{th}$ and further weakened into a cyclonic storm

in the morning hours (0830 IST) of the same day. It weakened into a deep depression and recurved its path towards north-northeastwards in the afternoon hours (1430 IST) over the south of Andhra Pradesh and further into a depression in the same midnight (2330 IST) over south coastal Andhra Pradesh. The observed track (Knapp et al., 2010) and intensity based on Dvorak classification (Dvorak, 1984) of NIVAR during $22^{nd}$ and $26^{th}$ November 2020 are shown in Fig. 1. NIVAR produced 130 mm of rainfall at Gadanki (13.5°N and 79.2°E) on $25^{th}$ and $26^{th}$, where the disdrometers observations were made. NIVAR passed

near Gadanki in the deep depression stage between 1430 and 1730 IST on the $26^{th}$.

A TC consists of a rain-free eye surrounded by a quasi-circular precipitation ring called an eyewall ($< 75$ km in radius) and spiral rainbands (Cecil et al., 2002). The spiral rainbands are further classified into inner (between 75 and 150 km) and outer ($> 150$ km) rainbands (Cecil et al., 2002). These regions are noted with concentric circles on the integrated multi-satellite retrievals for GPM (IMERG) final run V06B 30-minute rainfall (Huffman et al., 2020) spatial maps during $25^{th}$ and $26^{th}$ November 2020 (Fig. 2). Also shown in Fig. 2 are the NIVAR eye location indicated with a dot symbol and the Gadanki location with a star symbol. Over Gadanki region, NIVAR eyewall is produced rainfall during 1300 IST and 1600 IST on $26^{th}$, inner rainband between 0300 IST and 1300 IST, and after 1600 IST on $26^{th}$, and outer rainband during $25^{th}$ and up to 0300 IST on $26^{th}$. At Gadanki, tipping bucket rain gauge measurements show that the amount of rainfall produced by the NIVAR eyewall is 21 mm, the inner rainband is 83 mm, and the outer rainband is 26 mm.

The temporal variation of rain integral parameters (R, Z, and $D_m$) estimated from JWD, PARSIVEL, and LPM during the passage of NIVAR is shown in Fig. 3. The time series of R, Z, and $D_m$ shows a maximum of 38 mm h$^{-1}$, 44 dBZ, and 2 mm (except at once instant by LPM, which shows 2.5 mm), respectively. NIVAR's intensity and reflectivity observations are similar to the TC NISHA (formed during $24^{th}$ and $28^{th}$ November 2008 over the Bay of Bengal) observations at Gadanki (Radhakrishna and Rao, 2010). The $D_m$ observed during NIVAR is similar to the $D_m$ reported in cyclones elsewhere (Tokay et al., 2008; Wen et al., 2018) and in India (Radhakrishna and Rao, 2010). The rainfall observed during the passage of the outer rainband is mostly stratiform (rarely R $\geq 10$ mm h$^{-1}$), while in inner rainband and eyewall are both convective and stratiform in nature. The horizontal wind at 8 m height shows maximum speeds during the inner rainband and eyewall passage. The three disdrometers observed similar variations in rain integral parameters with time while showing differences in magnitudes due to variations in the measuring principle and hardware processing. The time series of 1-minute N(D) is plotted in Fig. 4 to investigate the differences in DSD observed by the three disdrometers. Irrespective of rain intensity, JWD rarely recorded raindrops greater than 3 mm, whereas LPM and PARSIVEL measurements showed raindrops up to 4 mm. The drops observed in the first few channels ($< 0.7$ mm) are relatively higher in LPM than in JWD and PARSIVEL. The overestimation of the number of drops by LPM is also noticed at other geophysical locations (Europe) by Angulo-Martínez et al. (2018) compared to PARSIVEL. As explained in Angulo-Martínez et al. (2018), although the measuring principle is the same for LPM and PARSIVEL, the differences seen in the DSD spectra could be due to differences in the laser beam dimensions that can count the splashes and margin fallers. However, the corrections done using theoretical fall velocity and sampling area removes these effects to a greater extent. Thus, the differences caused in the DSD spectra measured by the LPM and PARSIVEL could be due to variations in the hardware processing, which are undisclosed by the manufacturers.

The DSD differences observed between JWD, LPM, and PARSIVEL and their effect on rain integral parameters in different regions of NIVAR are studied using the variations of $D_m$, $N_w$, and Z with R. The slope and intercept of $D_m$-R curves estimated from JWD (red color), PARSIVEL (green color), and LPM (blue color) in the eyewall, inner, and outer rainband regions are shown in Fig. 5(a)-(c). For a given R, three disdrometers $D_m$ show comparatively largest values in the outer rainband, and larger values in the inner rainband than in the eyewall region. The three disdrometers $D_m$ increase with increasing R, while the magnitude of increase (slope) is different from each other. At a given R, the $D_m$ estimated from PARSIVEL is smaller, and JWD is larger than the other two disdrometers in all the regions of NIVAR. The DSD spectrum shape varies with R so that to make the

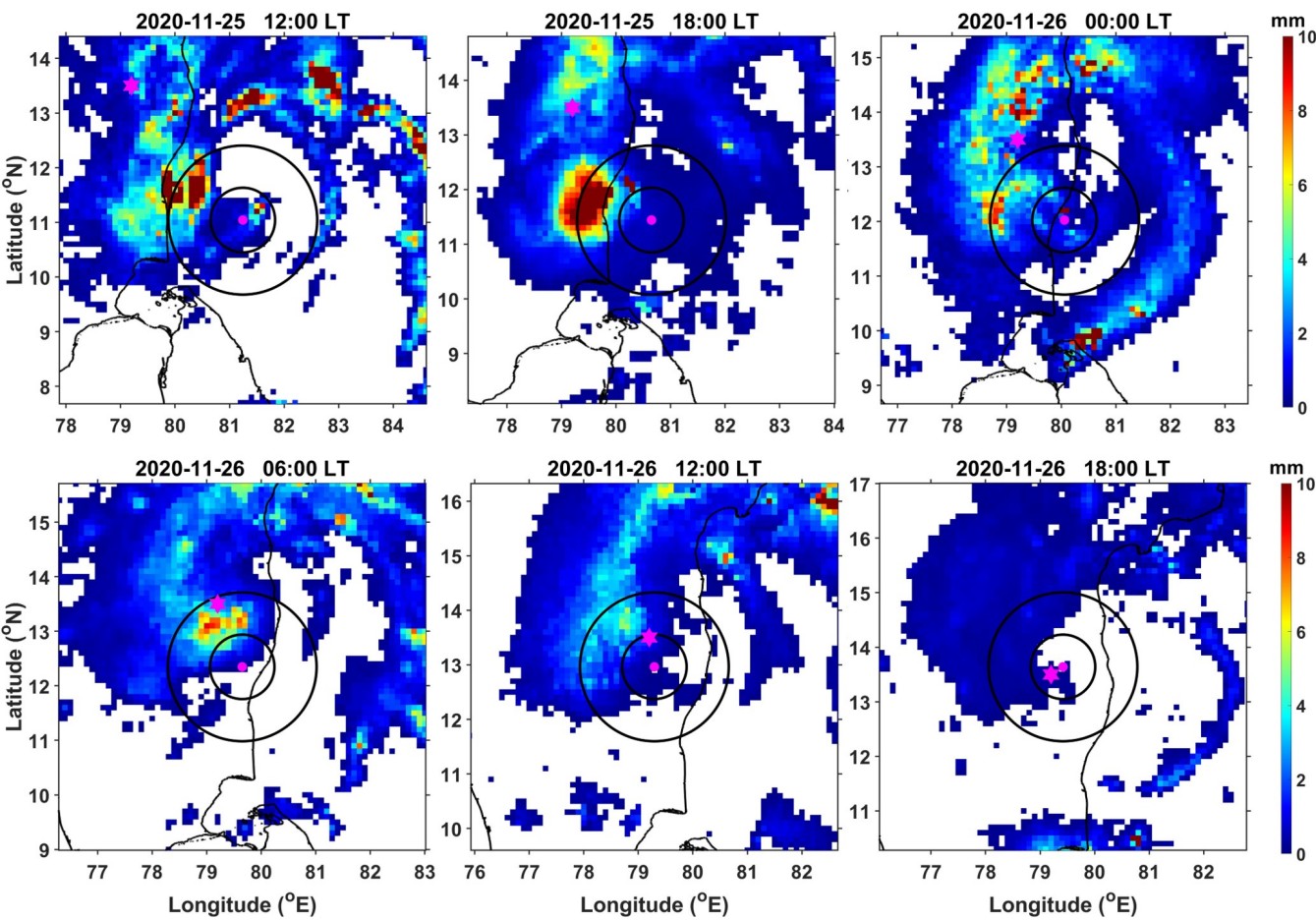

**Figure 2.** IMERG 30 minutes accumulated rainfall (in mm) maps of NIVAR cyclone with the eye (Pink closed circle), eyewall (75 km), and inner rainband (150 km) boundaries. The pink hexagon indicates the location of Gadanki, and the inner black solid circle show the eyewall/inner rainband and outer black solid circle indicate the inner/outer rainband boundaries.

spectra independent of shape, $N_w$ is considered following Testud et al. (2001). The variation of $10*log_{10}N_w$ (in dB where $N_w$ is in mm$^{-1}$ m$^{-3}$) with R in the three regions of NIVAR is depicted in Figs. 5(d)-(f). $N_w$ shows an increase with R in the outer rainband region for all disdrometers. In the eyewall region, JWD shows a decrease in $N_w$ with R (negative slope), where LPM and PARSIVEL show an increase (positive slope). During the passage of the inner rainband, JWD shows an increase in $N_w$ with R (positive slope), while LPM and PARSIVEL show a decrease (negative slopes). $N_w$-R curves show larger intercept values in the eyewall region and smaller values in the outer rainband region than in other regions. Nonetheless, the slope values vary for different regions and disdrometers. The discrepancy in the slopes of the $N_w$-R curves between the disdrometer in the eyewall and inner rainband needs to be further validated with more data before interpreting microphysically. Conventional weather radars use Z-R relations (Z=A*R$^b$, where A and b are empirical constants) for the quantitative precipitation estimation. The

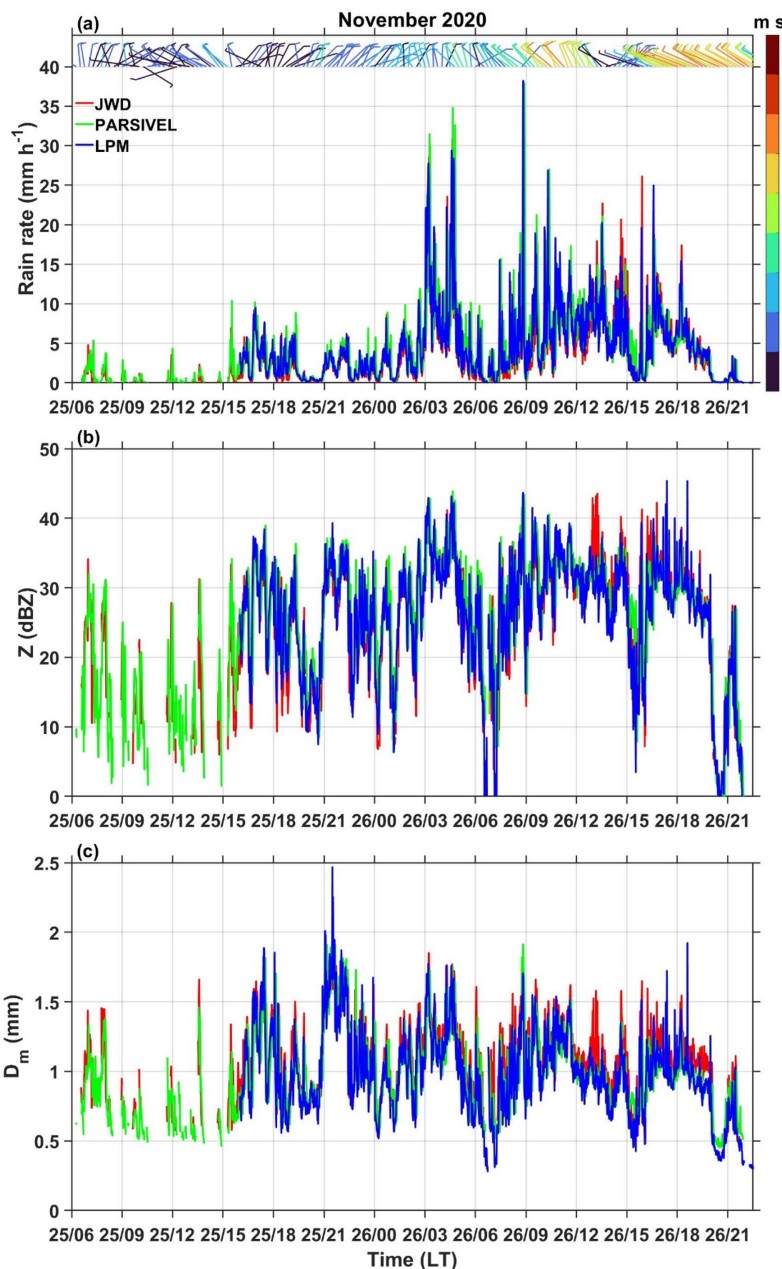

**Figure 3.** (a) Rain rate (R in mm h$^{-1}$), (b) reflectivity (Z in dBZ), and (c) mass-weighted mean diameter ($D_m$ in mm) observed by three kinds of disdrometers (JWD, PARSIVEL, and LPM) with a temporal resolution of 1 minute during the passage of NIVAR cyclone over Gadanki region. The wind barbs shown in (a) are the 5-min averaged wind vectors at 8 m height, whose magnitudes are indicated with the colors mentioned in the color bar.

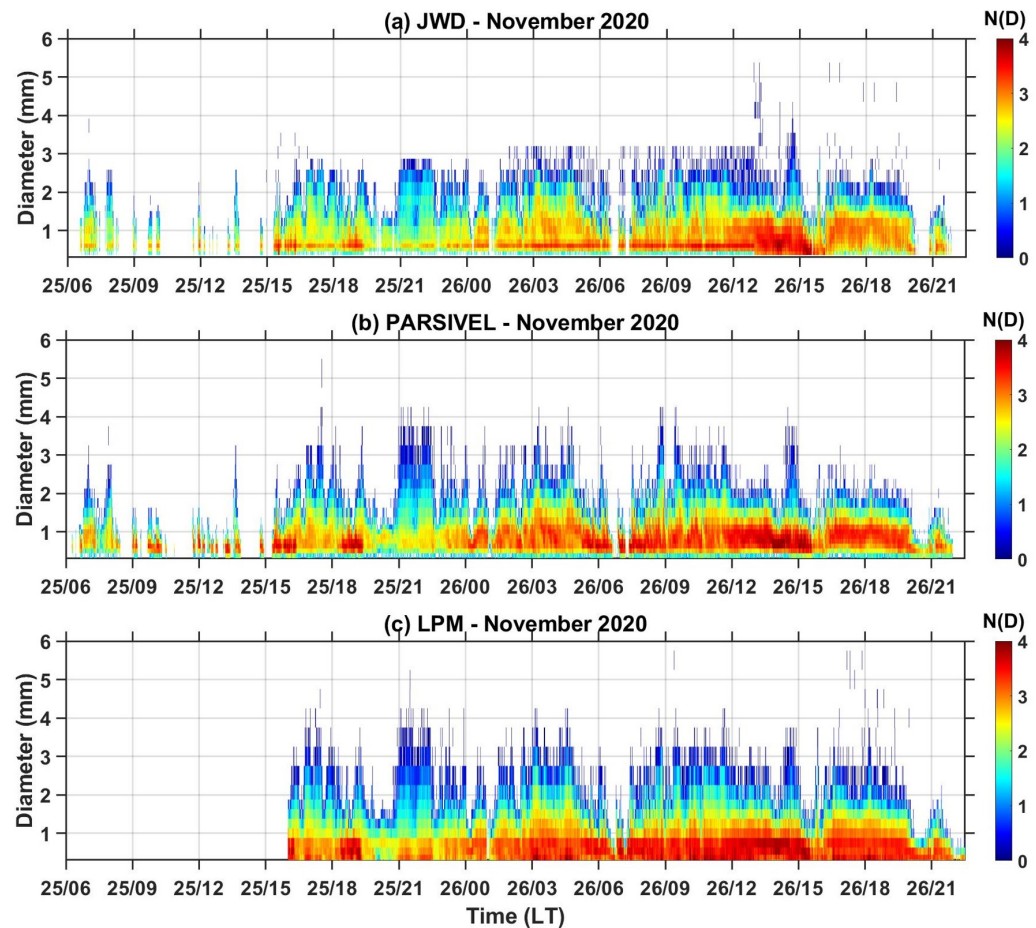

**Figure 4.** Time series of 1-minute N(D) in $mm^{-1}$ $m^{-3}$ observed by three kinds of disdrometers (JWD, PARSIVEL, and LPM) during the passage of NIVAR cyclone over Gadanki region. The colorbar indicates $log_{10}N(D)$.

Z-R relations are estimated using linear fit in log space and converted into power law forms for different disdrometers in various regions of a TC are depicted in Fig. 5(g)-(i) . Both empirical constants vary considerably between eyewall and other regions, suggesting that these regions' Z-R relations are distinctly different. The empirical coefficients vary from one disdrometer to another except for laser disdrometers in the outer rainband region. Comparing A and b values of the Z-R relations for a particular rain type in different regions provides information on precipitation microphysics. In particular. the coefficient A

gives the size of raindrops, i.e., larger A for larger raindrops, and the power b provides the microphysical processes. The size or mixed controlled cases where collision-coalescence dominates the b value is greater than one and for number-controlled case (collision, coalescence, and breakup) that produces equilibrium DSD b value is $\sim 1$ (Atlas et al., 1999; Rosenfeld and Ulbrich, 2003). The smaller A value of LPM than PARSIVEL and JWD in all the regions indicates the overestimation of smaller drops by LPM. The retrieved b value is greater than one by all disdrometers in all the regions, suggesting the dominance of the collision-

coalescence process. The Z-R relations obtained over the Atlantic basin during hurricane Anita (eyewall:$Z = 253R^{1.3}$; outer rainband:$Z = 341R^{1.25}$) are given in Marks et al. (1993) and over the east pacific basin during typhoon Lekima (Eyewall:$Z = 961.54R^{1.85}$; Inner rainband:$Z = 280.23R^{1.86}$; outer rainband:$Z = 74.25R^{1.98}$) in Bao et al. (2020), is distinctly different from the Bay of Bengal region (present study). Tropical cyclones over the Bay of Bengal and the Atlantic Ocean show an increase in A value and in turn the size of raindrops and $D_m$ with increasing distance from the cyclone eye while showing the
opposite in the eastern Pacific basin.

## 4 Effect of wind speed on estimated rain integral and polarimetric parameters

The vertical wind at aloft can influence the fall velocity of the hydrometeors. The vertical wind greater than $2\ m\ s^{-1}$ sustains very tiny time below 300 m altitude and persist for longer times at higher altitudes (Rogers et al., 1993). Thus, the vertical wind close to the earth surface is assumed to be small and its effect on drop fall velocity is neglected as raindrops of 4 mm
and large require less than 12 m to attain the terminal velocity (Van Boxel et al., 1997). As the disdrometers are installed at the earth's surface, the vertical wind effects are not considered in this study. The horizontal wind changes the raindrops falling path, resulting in variations in the recorded DSD spectrum. When a raindrop falls with an angle, the residence time of the raindrop in the laser beam increases, which enhances the attenuation at the detector, increases the measuring diameter, and decreases the fall velocity. The wind speed measured at 8 m altitude near the disdrometer location is considered to account
for the effects of horizontal wind on DSD measurements. The number of data points observed in the eyewall (8 one-minute samples) and outer rainband (6 one-minute samples) with wind speeds greater than 4 m s$^{-1}$ is small, so the present study is confined to two different wind speed intervals (0-2 m s$^{-1}$ and $> 2$ m s$^{-1}$).

The cyclonic DSDs are different from eyewall to inner rainband to outer rainband (Homeyer et al., 2021). The DSD observations during the NIVAR passage are first categorized into eyewall, inner, and outer rainbands following the classification
in Cecil et al. (2002). These categorized DSD spectra are further segregated with respect to R and wind speed, and the mean spectra are plotted in Fig. 6. The DSD observations are not available at R $> 10$ mm h$^{-1}$ with wind speed greater than 2 m s$^{-1}$ in the outer rainband, so the mean DSD spectra are not shown in Fig. 6. Since the observations are made at the same location, similarities between the three disdrometers specify the DSD characteristics of a TC, and disparities indicate the errors in the observations due to differences in the measuring principle and hardware processing of disdrometers. Similarities show
an increase in the maximum raindrop size with increasing R up to 5 mm h$^{-1}$, and at higher intensities, the slope of the DSD spectrum changes by increasing the number concentration of medium-sized raindrops (between 0.7 mm and 2 mm) at all wind speeds in the three regions of a TC. The disparities show overestimation of small raindrops ($< 0.7$ mm) by a factor of 10 to 100 by the LPM than JWD (except in the eyewall at R $< 5$ mm h$^{-1}$) and PARSIVEL at all R. At large drop end ($> 2$ mm), LPM and PARSIVEL overestimates raindrops concentration than JWD at R $> 5$ mm h$^{-1}$ in the inner rainband and at R $> 2$
mm h$^{-1}$ in the outer rainband while this overestimation is not seen in the eyewall region. The overestimation of large raindrops by laser disdrometers than JWD is not uniform in all the regions of NIVAR. This could be due to variations in the path of the falling raindrops from the vertical direction that cause errors in the measuring diameter of raindrops by the laser disdrometers

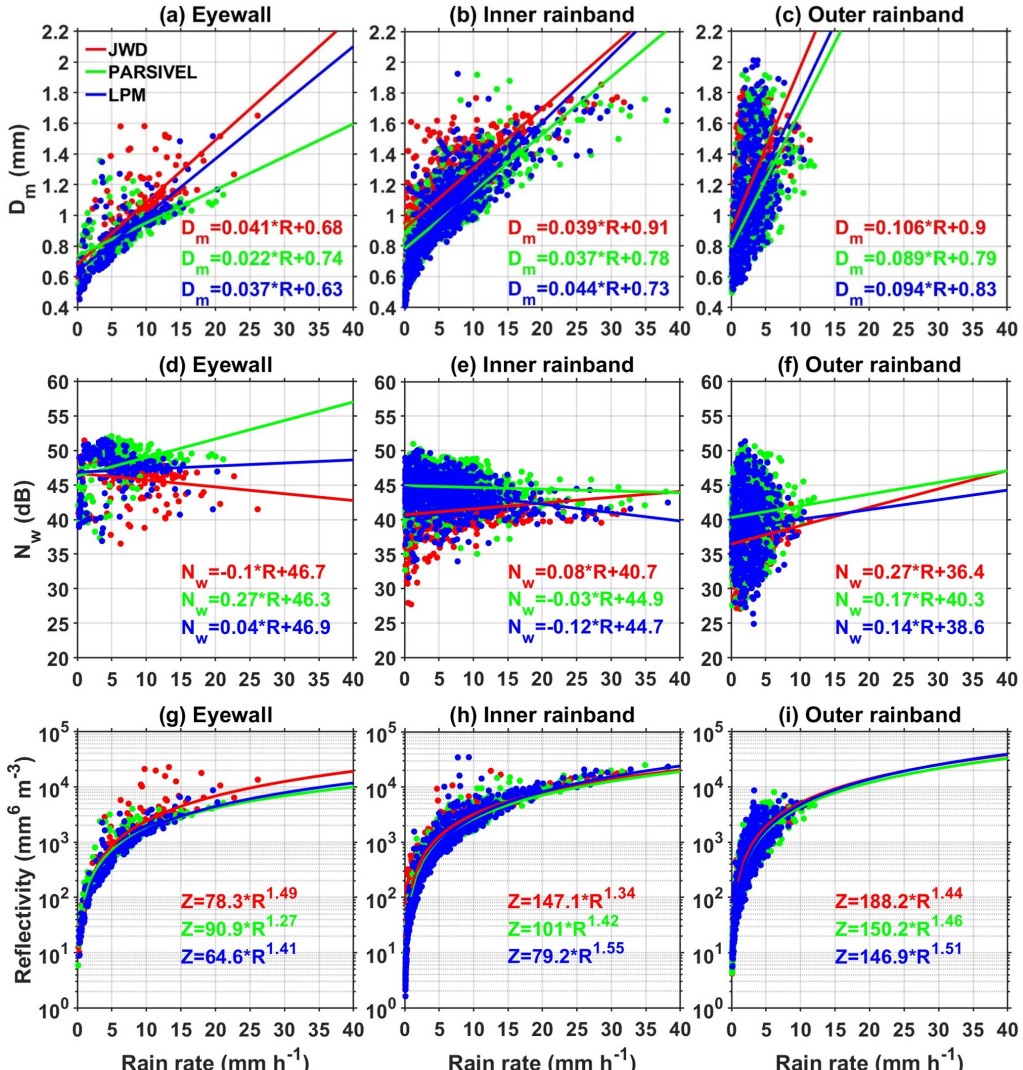

**Figure 5.** $D_m$ (mm) as a function of rain rate (mm h$^{-1}$) in (a) eyewall, (b) inner, and (c) outer rainband regions of NIVAR observed by JWD, PARSIVEL, and LPM. The solid lines represent the linear fit at 95% confidence level. (d)-(f) same as (a)-(c) but for $N_w$. (g)-(i) same as (a)-(c) but for Z and the solid lines represent the power-law fit whose relations are shown in legends with the respective color.

or hardware issues present in the JWD, as noted in Tokay et al. (2005). Compared to PARSIVEL, LPM records a marginally more number of larger drops ($> 2$ mm) could be due to changes in the hardware processing of these disdrometers.

The $D_m$-R data segregated based on wind speed and region of a TC NIVAR are depicted in Fig. 7. The best liner fit at 95% confidence level to the $D_m$-R data obtained from each disdrometer is also indicated with solid lines (JWD - red; PARSIVEL - green; LPM - blue) in Fig. 7. The effect of wind speed is not uniform for all the disdrometers in different regions of a TC. For

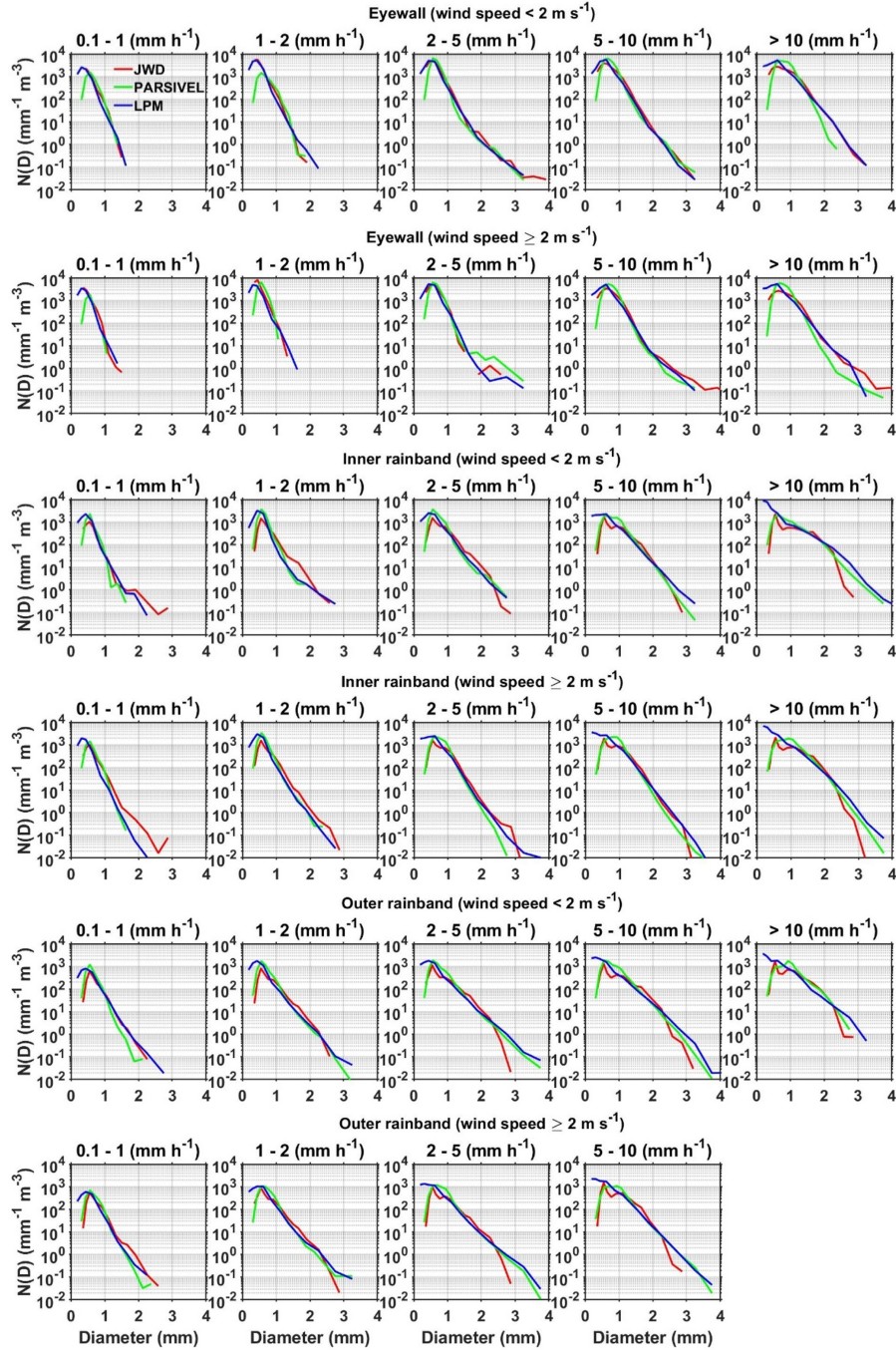

**Figure 6.** N(D) in mm$^{-1}$ m$^{-3}$ as a function of raindrop diameter (mm) in different rain rate and wind speed intervals associated with eyewall, inner rainband, and outer rainband of NIVAR cyclone observed by JWD, PARSIVEL, and LPM installed at Gadanki.

a given R, JWD shows an increase in $D_m$ with wind speed in the eyewall region, while small variation in $D_m$ with the wind in the inner and outer rainbands. PARSIVEL data show an increase in $D_m$ with the wind in the eyewall, a decrease in $D_m$ with the wind in the inner rainband, and small variations in the outer rainband. LPM shows an increase in $D_m$ with the wind in the eyewall and inner rainband and minor variations in the outer rainband. The observed differences in the $D_m$-R relations under the same environmental conditions indicate that the DSD spectra recorded by three disdrometers are different. At a giver R, irrespective of wind speed, large $D_m$ values are found in the outer rainband and small $D_m$ values in the eyewall region than in other regions of a TC. This is due to a decrease in the concentration of small raindrops and increases in the large raindrops from eyewall to inner rainband to outer rainband (Fig. 6). Though PARSIVEL underestimates the smaller drop concentrations, the estimated small $D_m$ values than LPM and JWD at all wind speeds with R > 5 mm h$^{-1}$ in the eyewall is due to recording a low concentration of large raindrops. At R < 5 mm h$^{-1}$, PARSIVEL recordings show the similar DSD distribution with LPM and JWD at medium to large raindrops with a low concentration of small raindrops, resulting in larger $D_m$. In the eyewall, the overestimate of small and underestimate of large raindrops by LPM than JWD results in relatively smaller $D_m$ values of LPM than JWD. However, at low wind speeds, the DSD distributions are similar, except the overestimation of small raindrops by LPM than JWD results in marginally small $D_m$ for LPM than JWD in the eyewall. In the inner rainband, the concentration of small raindrops observed by JWD and PARSIVEL are similar and low compared to LPM. At medium and large raindrops, the raindrop concentration observed by PARSIVEL and LPM is similar and lower than the JWD. Thus, at all wind speeds with R < 5 mm h$^{-1}$, the $D_m$ values are small for PARSIVEL and large for JWD in the inner rainband. At higher rain intensities, LPM overestimates the small raindrop concentration (by two orders of magnitude), while both LPM and PARSIVEL underestimates the medium-sized and overestimates the large-sized raindrops than JWD. The imbalance between the small, medium, and large raindrops results in large $D_m$ values for JWD at all wind speeds, while for LPM small $D_m$ values at wind speed less than 2 m s$^{-1}$, and large $D_m$ values at higher wind speeds than for PARSIVEL in the inner rainband. Although LPM and PARSIVEL show nearly the same distribution at the medium raindrops in the outer rainband, LPM overestimates the small and large raindrops, resulting in marginally larger $D_m$ than PARSIVEL at all R and wind. Compared to JWD, LPM and PARSIVEL records a high concentration of large raindrops and a low concentration of medium-sized raindrops at all R and wind, which imbalance the DSD spectrum to produce marginally small $D_m$ than JWD in the outer rainband.

The normalized DSD (Testud et al., 2001) indicates $N_w$ (mm$^{-1}$ m$^{-3}$) is an intercept parameter of the exponential DSD with the same liquid water content and $D_m$ of an observed DSD spectrum with any shape. $N_w$ is converted into dB ($10 * log_{10} N_w$) and as a function of R and wind for different regions of NIVAR are plotted in Fig. 8 to understand the effect of wind on drop concentration. In general, $N_w$ increases with increasing R (Testud et al., 2001), while this is not always true when there is an imbalance between the decrease in small and increase in medium and large size raindrops (Ma et al., 2019). The $N_w$-R curves are different for various regions of a TC and varies with wind speed. JWD shows an increase in $N_w$ with R in the inner and outer rainbands while a decrease in the eyewall at all wind speeds. The decrease in $N_w$ with R of JWD is small at lower wind speed and considerable at higher wind speeds. PARSIVEL measurements indicate an increase in $N_w$ with R in the eyewall and outer rainbands while a decrease in the inner rainband. The change in $N_w$ with R of PARSIVEL is considerable at all wind speed in all the regions of a TC except at low wind speeds in the outer rainband and high wind speeds in the inner rainband.

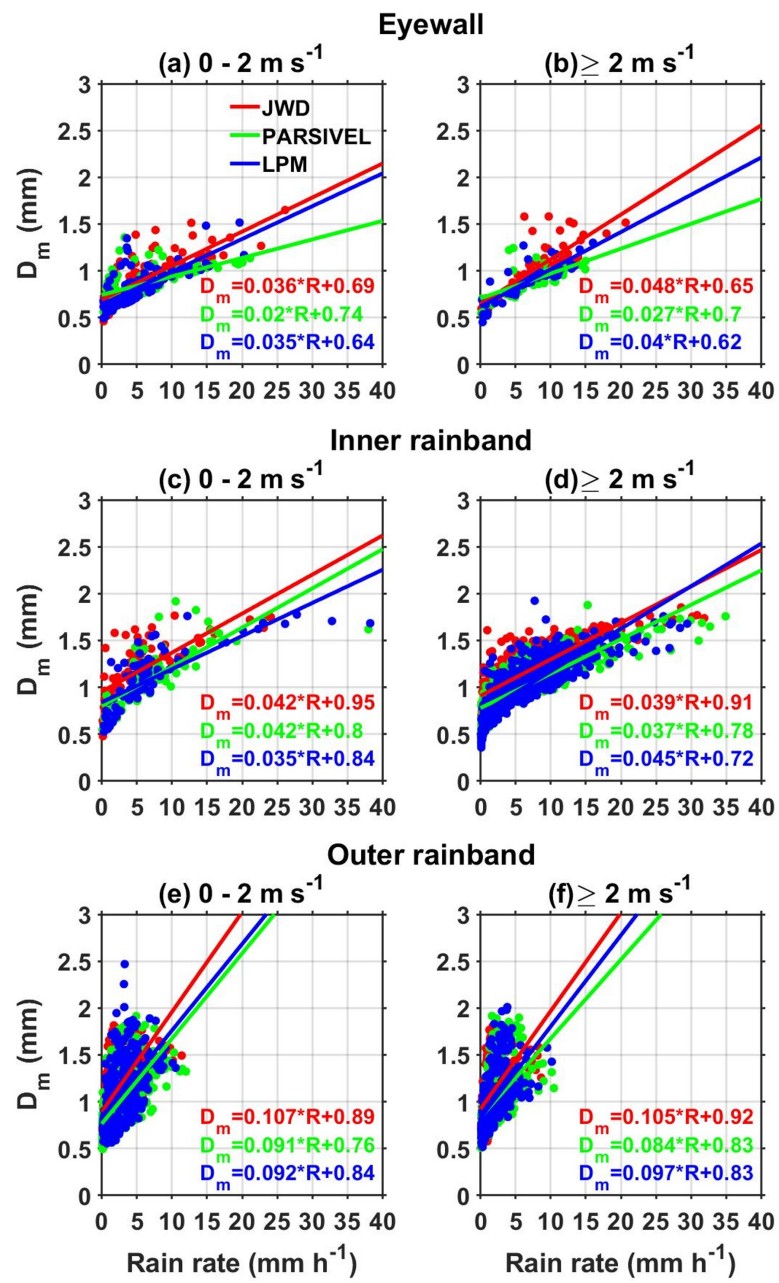

**Figure 7.** (a)-(b) $D_m$ (mm) as a function of rain rate (mm h-1) in the eyewall of NIVAR observed by JWD, PARSIVEL, and LPM during different surface wind speed intervals. The solid lines represent the linear fit at 95% confidence level whose relations are shown in legends with the respective color. (c)-(d) and (e)-(f) are the same as (a)-(b) but in the inner and outer rainbands of NIVAR, respectively.

LPM data show an increase in $N_w$ with R in the outer rainband and a decrease in the inner rainband while increasing at low wind speeds and decreasing at high wind speeds in the eyewall. A sizable change in $N_w$ with R of LPM is observed in the
inner rainband and at high winds in the outer rainband and small in the eyewall and low windspeeds in the outer rainband. The $N_w$ values are larger for PARSIVEL than LPM in three regions of a TC at all wind speeds. This could be due to the presence of more large drops in LPM than PARSIVEL. JWD shows smaller $N_w$ values than LPM and PARSIVEL at R less than 15 mm h$^{-1}$. The change in the concentration of small raindrops observed by three disdrometers with R and wind speed is minimal in the outer rainband, resulting in an increase in $N_w$ with R, as also observed in Figs. 5. Nonetheless, in the inner rainband
and eyewall, the small drop concentration increases with R at all wind speeds, making an imbalance between the small and medium-sized raindrops that cause variations (increase/decrease) in $N_w$ with R differently for different disdrometers.

The polarimetric parameter $Z_{DR}$ provides information on measuring the reflectivity-weighted hydrometeors' shape within a sampling volume. $Z_{DR}$ at a temperature of 20 °C (average surface temperature is 21 °C at Gadanki during the passage of NIVAR) in the X-band frequency estimated from the DSD spectra of JWD, LPM, and PARSIVEL as a function of R at
different wind speeds are depicted in Fig. 9. For a given R, all disdrometers show large $Z_{DR}$ in the outer rainband than in other regions of a TC. Relating three disdrometers, LPM shows large values than PARSIVEL and JWD in all regions of NIVAR except at wind speeds greater than 2 m s$^{-1}$ in the eyewall, where JWD shows relatively large values. These observations are in accordance with the measure of more large raindrops by LPM in all the regions except in the eyewall at high wind speeds. Though $D_m$ values of JWD are large and LPM are small in all regions, the small $Z_{DR}$ derived from JWD and large
$Z_{DR}$ from LPM indicate the dependency of large raindrops is more pronounced in computing $Z_{DR}$ than $D_m$. This could be due to the resonance effect of raindrops with drops greater than 3 mm in diameter at X-band frequency (Carey and Petersen, 2015) as depicted in Fig. 10. $Z_{DR}$ at S-band show monotonic behaviour with raindrop diameter while at C- and X-bands show nonmonotonic behaviour. The nonmonotonic behaviour is mainly due to the resonance effect at D > 5 mm for C-band and D > 3 mm for X-band frequency radars. At resonating frequencies, the maximum deviation in $Z_{DR}$ between C-band and S-band
is ∼5 dB, between X-band and S-band is ∼0.7 dB. As the maximum raindrop size observed during NIVAR is less than 4 mm, the resonance effect is not applicable for C- and S-band retrievals. Regardless of wind speed, the laser disdrometer shows a large $Z_{DR}$ in the inner rainband than in the eyewall, while JWD displays opposite features. This is due to the resonance effect caused by the presence of raindrops with a diameter greater than 3 mm in the eyewall region of the JWD data while in the inner rainband in the laser disdrometers data (Fig. 6). $Z_{DR}$ estimated from LPM are marginally larger at high wind speeds than at
low wind speeds in all regions of NIVAR. JWD estimated $Z_{DR}$ increase with wind speed in the eyewall and nearly the same in inner and outer rainbands. $Z_{DR}$ of PARSIVEL shows an increase with wind speed in the eyewall, a decrease in the inner rainband, and no change in the outer rainband.

The $K_{DP}$ offers information on the mass of nonspherical hydrometeors in the volume of a radar beam. The $K_{DP}$ estimated at X-band frequency with a temperature of 20 °C from three disdrometers as a function of R at different wind speeds is
depicted in Fig. 11. The power-law relations of $K_{DP}$-R are also shown in Fig. 11. The $K_{DP}$-R relations show diversity in different regions of NIVAR, but all disdrometers show approximately the same relations in a given region except PARSIVEL

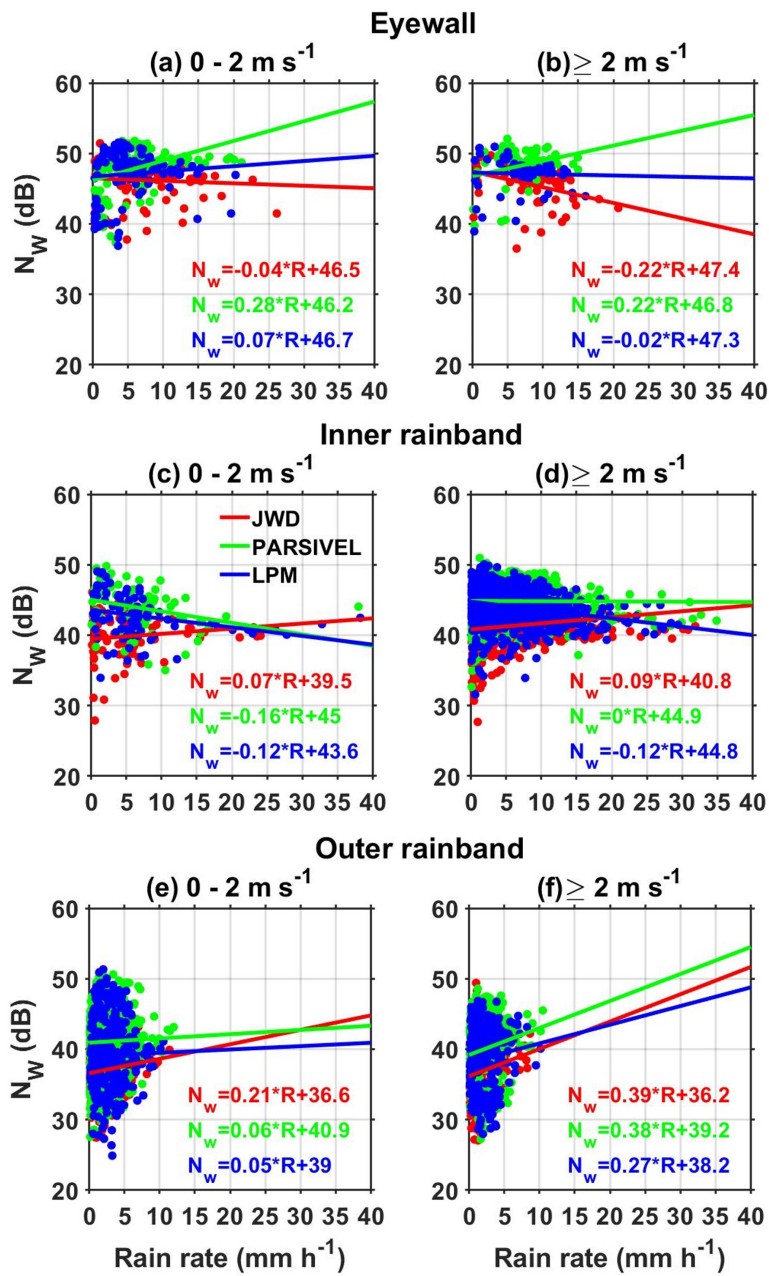

**Figure 8.** Same as Fig. 7 but for $N_w$ (dB).

in the eyewall. The increase in $K_{DP}$ indicates the increase in nonspherical particles with wind speed in the eyewall. However, $K_{DP}$ decreases with wind speed in the outer rainband and shows the same values in the inner rainband.

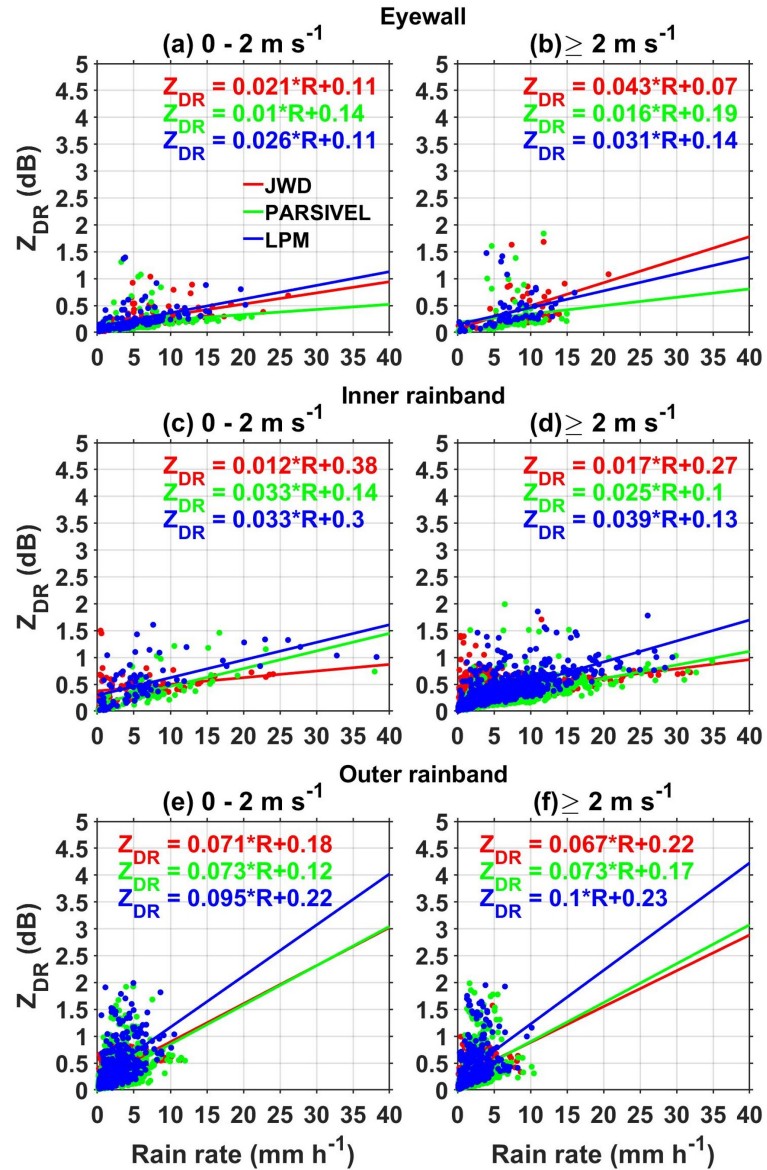

**Figure 9.** (a)-(b) $Z_{DR}$ (dB) as a function of rain rate (mm h-1) in the eyewall of NIVAR observed by JWD, PARSIVEL, and LPM during different surface wind speed intervals at X-band frequency in the ambient atmosphere with 20 $^\circ$C temperature. The solid lines represent the linear fit at 95% confidence level whose relations are shown in legends with the respective color. (c)-(d) and (e)-(f) are the same as (a)-(b) but in the inner and outer rainbands of NIVAR, respectively.

The polarimetric parameter $K_{DP}$ is measured using the phase difference between the two polarizations, which is immune

from attenuation. Hence $K_{DP}$ is widely used to correct the attenuation and differential attenuation. Molecular absorption

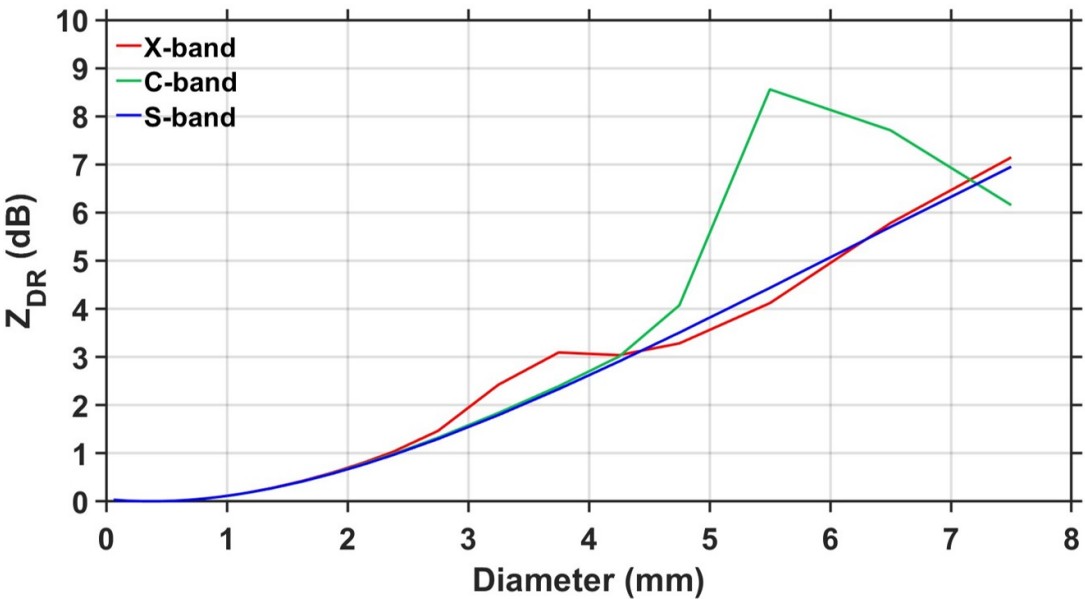

**Figure 10.** $Z_{DR}$ (dB) as a function of monodisperse raindrop diameter (mm) at X-band (red), C-band (green), and S-band (blue) wavelengths. For the monodisperse simulations at a drop temperature of 20 °C, the refractive index of raindrops is estimated from Ray (1972), drop axis ratio is considered from Brandes et al. (2002).

and scattering out of the beam control the attenuation. The relations between $A_H$, $A_{DP}$, and $K_{DP}$ are given in (17) and (18), and DSD measurements obtained from disdrometers are used to estimate these relations, whose coefficients are reliant on temperature (Jameson, 1992). The coefficient $\gamma_H$ of $A_H$-$K_{DP}$ relation at X-band frequency in the eyewall, inner, and outer rainbands at different temperatures and wind speeds are plotted in Fig. 12. The $\gamma_H$ estimated from JWD, LPM, and
PARSIVEL decrease with increasing temperature in all the regions of a TC. The molecular absorption (the imaginary part of the complex refractive index) enhances at low temperatures and causes an increase in attenuation with a decrease in temperature (Jameson, 1992; Smyth and Illingworth, 1998). The change in $\gamma_H$ with temperature is not uniform in different regions of NIVAR due to variations in DSD between eyewall, inner, and outer rain bands. $\gamma_H$ estimated from JWD is small (except in the eyewall at temperature $> 25$ °C and wind speed $> 2$ m s$^{-1}$) than other disdrometers in all regions of NIVAR. LPM
estimated $\gamma_H$ values are larger in the inner and outer rainbands while PARSIVEL in the eyewall at all temperatures and wind speeds than other disdrometers. In the inner and outer rainbands, the derived $\gamma_H$ values are the same for laser disdrometers at lower temperatures and show marginal differences with increasing temperature. For a given temperature, $\gamma_H$ derived from all disdrometers show slightly larger values at high wind speeds than at low wind speeds. For a given temperature and wind, $\gamma_H$ shows negligible variations within the regions of a TC except for PARSIVEL in the eyewall. The $\gamma_H$ variation with temperature
shows considerable difference between the laser disdrometers in eyewall region at all wind speeds due to large differences in $k_{DP}$ between them.

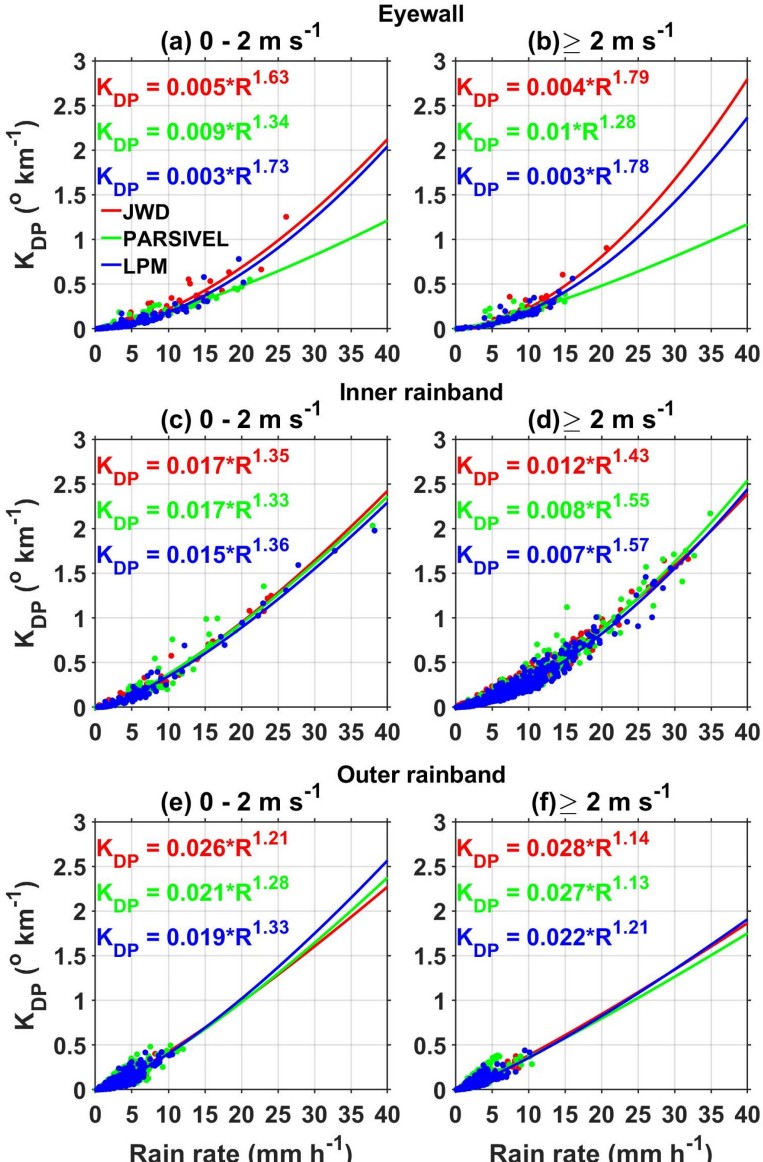

**Figure 11.** Same as Fig. 9 but for $K_{DP}$ ($^\circ$ km$^{-1}$) and the solid lines represent the power-law fit.

The differential attenuation coefficient $\gamma_{DP}$ derived from $A_{DP}$-$K_{DP}$ relations from JWD, LPM, and PARSIVEL at different temperatures and wind in the eyewall, inner, and outer rainbands are depicted in Fig. 13. For a given wind and temperature, large $\gamma_{DP}$ values are observed for LPM and small values for JWD than other disdrometers in the inner and outer rainbands. In the eyewall for a given wind and temperature, $\gamma_{DP}$ values are smaller for PARSIVEL, larger for LPM at wind speeds less than 2 m s$^{-1}$, and JWD at higher wind speeds than other disdrometers. JWD estimated $\gamma_{DP}$ values show a small decrease with an

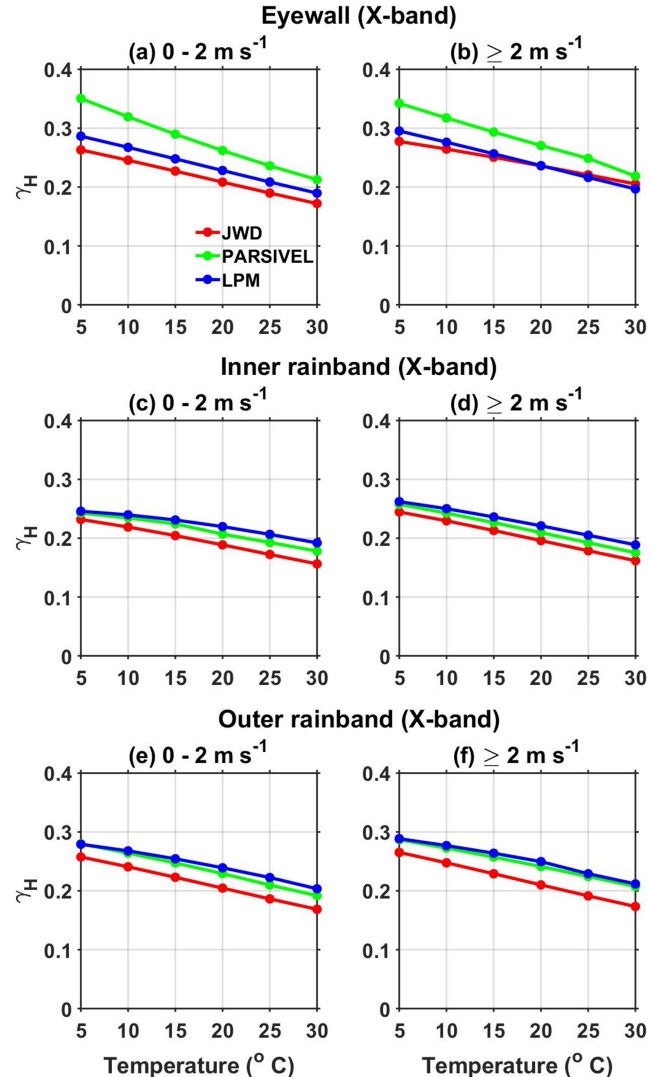

**Figure 12.** (a)-(b) $\gamma_H$ as a function of temperature ($^\circ$C) in the eyewall of NIVAR observed by JWD, PARSIVEL, and LPM during different surface wind speed intervals at X-band frequency. (c)-(d) and (e)-(f) are the same as (a)-(b) but for the inner and outer rainbands of NIVAR, respectively.

increase in temperature in all the regions of a TC (except eyewall at high wind speeds). LPM and PARSIVEL estimated $\gamma_{DP}$ values show a minuscule decrease with an increase in temperature in all regions of NIVAR at all wind speeds. JWD estimated $\gamma_{DP}$ values are larger at high wind speeds than at low wind speeds in the eyewall and do not show variations with wind speeds in the inner and outer rainbands. PARSIVEL and LPM estimated $\gamma_{DP}$ values are larger or nearly equal at high wind speeds than at low wind speeds in the eyewall and outer rainband while smaller in the inner rainband. Comparing Figs. 12 and 13, the


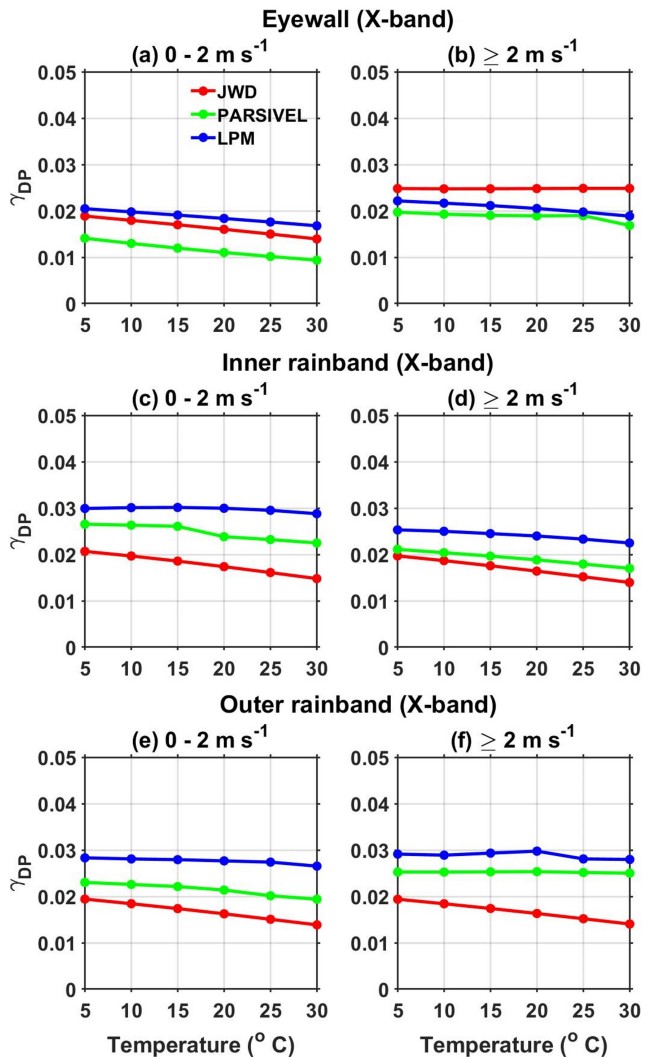

**Figure 13.** Same as Fig. 11 but for $\gamma_{DP}$.

$\gamma_{DP}$ values are smaller and $\gamma_H$ values are larger for PARSIVEL than LPM in the eyewall region indicating the effect of more medium sized raindrops on $Z_{DR}$, $K_{DP}$, $A_H$, and $A_{DP}$. Presence of more number of medium sized raindrops that are slightly deviated from sphericity will result in smaller $Z_{DR}$, $K_{DP}$, and $A_{DP}$ and larger $A_H$.

The variation of $\gamma_H$ and $\gamma_{DP}$ with temperature estimated from DSD data recordings of JWD, LPM, and PARSIVEL at C-band and S-band frequencies are depicted in Figs. 1S-4S. Similar to the earlier studies (Bringi et al., 1990; Jameson, 1991; Park et al., 2005), the $\gamma_H$ and $\gamma_{DP}$ values are smaller at S-band followed by C-band than at X-band. All disdrometers estimated $\gamma_H$ and $\gamma_{DP}$ values decrease with increasing temperature both at S- and C-bands. For a given temperature and wind, $\gamma_H$ values are approximately the same for the three disdrometers in the inner and outer rainbands but show differences in the eyewall at S-

and C-bands. Also, the effect of wind on $\gamma_H$ is negligible in the eyewall and inner rainbands. Similar to X-band frequency, the $\gamma_H$ values estimated at S- and C-band also show larger values for PARSIVEL and smaller for JWD than for other disdrometers in the eyewall regardless of the temperature and wind. Unlike $\gamma_H$, $\gamma_{DP}$ shows variations between disdrometer in any given region of NIVAR at C-band while showing negligible variations at S-band. Regardless of temperature and wind $\gamma_{DP}$ values are the same for a given disdrometer in the inner and outer rainbands but increase with wind speed in the eyewall.

## 5    Conclusions


The characteristics of landfalling TC NIVAR are revealed using JWD, PARSIVEL, and LPM observations made at Gadanki, India. The three disdrometers are installed at the same location; the measurements are used to study the effect of wind speed and variations in measuring principles and data processing algorithms on the recorded DSD spectra and, in turn, on the retrieved rain integral and polarimetric parameters.

1. JWD measures raindrops of diameters up to 3 mm while LPM and PARSIVEL record up to 4 mm. The raindrop more residing time in the laser beam due to deviation in fall path from nadir by strong horizontal winds resulting in an additional reduction in the beam intensity at the receiver. Thus, the laser disdrometers overestimate the size of the raindrops in the presence of horizontal winds.

2. The DSD spectrum width increases with increasing R by observing larger-sized raindrops. Also, the concentration of
raindrops of diameters between 0.7 mm to 1.5 mm increases in all the regions of a TC. However, the magnitude of the increase is high in the eyewall than in the inner and outer rainbands.

3. The DSD characteristics reveal relatively larger $D_m$ in the outer rainband and smaller $D_m$ in the eyewall than in other regions of a TC. The maximum $D_m$ observed is less than 2 mm, which follows the earlier studies. Raindrops of diameter 3 mm in size are observed infrequently in the eyewall, while they are present in the inner and outer rainbands at R greater
than 5 mm h$^{-1}$.

4. The Z-R relations are distinctly different in various regions of a TC and for different disdrometers. The Z-R relations estimated from three disdrometers indicate comparatively larger Z for a given R in the outer rainband followed by the inner rainband and smaller Z in the eyewall.

5. The $N_w$ increases with increasing R at all wind speeds in the outer rainband while showing an increase/decrease dif-
ferently for various disdrometers in the eyewall and inner rainbands. The imbalance between small and medium-sized raindrops causes variations in $N_w$ with R at different wind speeds.

6. $Z_{DR}$ estimated at X-band frequency with a temperature of 20 °C shows larger values in the outer rainband than in the eyewall and inner rainband. Three disdrometers estimated $Z_{DR}$ show differences in inner rainband and eyewall at different wind speeds. In the inner and outer rainbands, the laser disdrometers observe raindrops with a diameter greater

than 3 mm, which cause resonance at X-band frequency and results in large $Z_{DR}$ than JWD, whose measures show raindrops till 3 mm only.

7. In the eyewall region, the observed smaller $K_{DP}$ by PARSIVEL at all wind speeds and R indicates the presence of a low number concentration of nonspherical raindrops results in smaller $Z_{DR}$ values than in LPM and JWD.

8. The coefficients of attenuation ($\gamma_H$) and specific attenuation ($\gamma_{DP}$) decrease with increasing temperature but differ for different disdrometers. Regardless of wind, for a given $K_{DP}$, attenuation and differential attenuation are more for LPM and PARSVEL than JWD in inner and outer rainbands while differing in the eyewall.

9. LPM overestimates the small raindrops (< 0.7 mm) by a factor of 10 to 100 than JWD (except in the eyewall) and PARSIVEL at all R. At the large drop end (> 2 mm), JWD underestimates raindrops concentration than LPM and PARSIVEL at R > 5 mm h$^{-1}$ in the inner rainband and at R > 2 mm h$^{-1}$ in the outer rainband while this underestimation is not seen in the eyewall region. The underestimation of large raindrops by JWD is not uniform in all the regions of NIVAR. Compared to PARSIVEL, LPM records a marginally more number of larger drops (> 2 mm).

10. The effect of wind speed on the recorded DSD and estimated rain integral and polarimetric parameters are not uniform in various regions of NIVAR for different disdrometers as these effects are further modified by measuring principle and hardware processing.

*Data availability.* The complete dataset used in the analysis can be obtained by contacting the data dissemination unit of the National Atmospheric Research Laboratory. The figures are generated using MATLAB software.

*Author contributions.* Basivi Radhakrishna conceived the idea, conducted the detailed analysis, and contributed to the writing.

*Competing interests.* The author declare that he has no conflict of interest.

*Acknowledgements.* The author thank the three anonymous reviewers and Associate editor Dr. Zamin A. Kanji for their suggestions in
improving the quality of the manuscript.

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
