# Peer review of "Raindrop Size Distribution (DSD) during the Passage of a Tropical Cyclone NIVAR: Effect of Measuring Principle and Wind on DSDs and Retrieved Rain Integral and Polarimetric Parameters from Impact and Laser Disdrometers"

_Atmospheric Measurement Techniques, 2022_

## Referee Comment (RC2)

**Review of AMT-2022-209**

**General comments**

This study shows a thorough analysis of DSD measurements from three co-located disdrometers of different types, located in Gadanki, India, during the landfall of cyclone NIVAR in November 2020. The spectra and values of key rainfall parameters are compared between the disdrometers, for different storm regions and wind speeds. The results provide a useful comparison between these three disdrometer types, and a novel aspect here is that the rainfall is not "typical" because NIVAR was a tropical cyclone.

While the comparison is thorough, it is disappointing that there is not more analysis of what the results mean in terms of the properties of the cyclone rainfall that has been sampled. The rich data source explored here would make for a useful comparison of the rainfall properties that are experienced in the different parts of the tropical cyclone - for example which parts are influenced most heavily by the drop concentration and which are influenced most heavily by drop size. Particularly concerning Figures 10, 11 and 12, the results are simply stated without physical explanations. A proper discussion of the results and their physical meanings, with references to the literature on tropical cyclone rain properties, is required. I have listed other recommendations below.

**General comments**

1. There are some grammatical errors which can affect the readability of the manuscript at times. These errors are often to do with comparisons: for example, on line 6: "high" should read "higher"; on line 7 "large" should read "larger", and so on throughout the paper. On line 61, the artefacts and errors themselves are not essential but taking them into account is essential. The use of tense in Section 3 is inconsistent. These errors are generally minor in nature and a thorough edit will fix them.

2. The introduction should include an introduction to what the raindrop size distribution is and its importance (to e.g. remote sensing and numerical weather prediction).

3. The results need to be put into more context with other studies. For example, the authors have found that different parts of the cyclone produced very different Z-R relationships. What have other authors found for cyclone Z-Rs and how do they compare to these results?

This lack of discussion extends to the other results and their physical meanings in terms of rainfall in tropical cyclones.

**Line-by-line comments**

1. Line 24: "Convective processes and resulting rainfall in a TC are primarily governed by the evolution of the microphysics of a TC." This statement needs a reference.

2. Line 35: On underestimation of small raindrops by disdrometers, Thurai et al. (2017) also reported on this underestimation and Raupach et al. (2019) proposed a possible solution.

3. Line 70: For the laser disdrometers was any filtering on fall velocity by drop size done, as in e.g. Jaffrain and Berne (2011)?

4. Line 75: $i$ must be the diameter interval number, not the number of intervals.

5. Equation 2: What do the numbers 4600 and 1000 in this fraction

6. Line 80: A reference for the LPM should be included. represent? They do not align with the given laser dimensions.

7. Line 93: Units should be provided for $v(j)$.

8. Line 97: A reference for the PARSIVEL disdrometer should be provided.

9. Line 100: In Equation 5, $D/2$ is often used (as stated here) and yet newer PARSIVEL disdrometers automatically remove any raindrop that touches the edge of the laser area; in this case the effective sampling area should be calculated using $D$ instead of $D/2$. The authors should check which is used in this case.

10. Equation 6: $v(j)$ should be properly defined here to show that it refers to the $j$th PARSIVEL velocity class.

11. Line 104: The locations (ie coordinates) of the disdrometers should be given, as well as their altitudes and the situation in which they are installed (e.g. open field, building roof, etc).

12. Lines 106-109: How are these thresholds decided; were they based on previous studies?

13. Line 111: It should be noted in the paper that this 6th-DSD-moment $Z$ is reflectivity in the Rayleigh regime, whereas the T-matrix calculations used later in the paper are in the Mie regime.

14. Line 111: $D_m$ should be labelled here as mass-weighted mean diameter.

15. Equation 11: I think the $\pi^4$ in this equation should be $\pi^5$; please double check.

16. Lines 140-141: The authors should reference attenuation-correction studies that use this technique here.

17. Equations 11-16: $\lambda$, $K$ should be defined with units and the meanings of *Re* and *Im* should be written out.

18. Equations 17 and 18: $\gamma_{DP}$ and $\gamma_H$ require definitions, and these equations require better explanation.

19. Figure 1: axis labels are missing; the Dvorak classification requires a reference on line 160; it should be stated what time interval is represented between each point that is plotted.

20. Line 164: This statement about the eyewall requires a reference.

21. Figure 2: Axes are missing labels, and the caption should state that the black solid lines show the inner/outer boundaries.

22. Line 170: What type of rain gauges were used and how close were they to the disdrometers?

23. Figure 3: It is important that the caption states the time resolution of the measurements shown here, since rain rate depends on resolution.

24. Line 174: The maximum $D_m$ is 2.5 mm – why do the authors discount the LPM measurement?

25. Figure 4: The 1-minute resolution should also be mentioned in the caption for this figure.

26. Line 184: Given other studies (e.g. Thurai et al. (2017)), it is possible that the PARSIVEL has underestimated the number of small drops rather than the LPM overestimating the numbers of drops.

27. Line 188: The authors mention corrections based on theoretical fall velocity, yet no corrections are mentioned in Section 2.

28. Figure 5: Are these linear fits statistically significant? The authors should show significance information and discuss.

29. Line 206: What method is used to fit the Z-R relations? If a linear relationship in log space is used it needs to be stated to distinguish the method from other methods that fit power laws specifically. The caption mentions a power-law fit but not which method was used.

30. Line 213: It's not clear here why vertical wind speed near the surface is insignificant – I would think that vertical wind strong enough to loft 4 mm drops is easily obtained both aloft and near the surface in convective storms.

31. Line 220: Exactly how many data points with wind over 4 m s$^{-1}$ were observed? From Figure 1 it appears that this number cannot be insignificant since there are large areas where the five-minute averaged wind speed was in the 5-8 m s$^{-1}$ range. Given that the event in question is a cyclone it seems reasonable that there may have been some strong winds that could skew the statistics for a ¿ 2 m s$^{-1}$ wind speed category. The authors should discuss this point and justify the categories used.

32. Line 222: It would be helpful to briefly explain the DSD classification used here.

33. Line 226: Rainfall variability is high enough that even "co-located" instruments metres apart sample different rainfall properties, so not all differences can be put down to instrument error or measurement principle.

34. Figure 6: It is possible that extreme values skew the mean DSDs shown here. The authors should test whether the median DSDs are very different - if they are, then showing the median DSDs may be more representative of the "characteristic" DSD.

35. Line 230: Again I wonder whether what the authors call an "overestimation" by the LPM is actually an underestimation by PARSIVEL and JWD?

36. Line 234: The different properties of JWD underestimation in different storm regions makes me wonder whether the physical set-up of the instrument could play a role - i.e. if there is a nearby building wind direction could make a difference.

37. Line 233: The JWD also records more large drops than the other instruments in the inner rainband for low rain rates.

38. Lines 240-245: It is unclear here how the authors are judging whether a difference in slope between the different plots is significant or not, and this should be stated. For example the difference between JWD lines in the eyewall is only slightly larger than differences in the other storm regions. The authors should also use language that acknowledges the uncertainty – ie use "little difference" instead of "no difference" when the differences are not significant, and "similar" instead of "same" distribution, since there are still differences present.

39. Line 252: Do the authors mean at wind speed $> 2$ m s$^{-1}$ instead of $R > 2$ mm h$^{-1}$?

40. Lines 255-265: The repetitive nature of the results shown here makes this section difficult to read and it is unclear whether the authors are referring to Figure 7 or Figure 6 or to both figures. The discussion could be made more concise.

41. Lines 259-261: This sentence reports on all wind speeds but also wind speeds less than 2 m s$^{-1}$ which does not make sense.

42. Line 263: The claim that LPM shows smaller $D_m$ than PARSIVEL in the outer rain band is not supported by Figure 7.

43. Line 267: $10 \log_{10}^{N_w}$ should be written $10 \log_{10} N_w$.

44. Line 269: The statement that "In general, $N_w$ increases with increasing $R$" is not supported by Figure 8. Do the authors mean that they expect $N_w$ to increase with $R$, given previous work?

45. Line 272: "$N_w$ is smaller in the eyewall while larger in the inner and outer rainbands than at lower wind speeds" - this statement contradicts Figure 8 which shows that $N_w$ is generally larger in the eyewall than in other storm regions. The authors meaning here is unclear.

46. Figure 8: The statistical significance of the linear fits should be discussed. It may well be that some of these fits are not significant enough to show an increase or decrease of $N_w$ with

$R$, since by eye the slopes often look close to zero.

47. Line 277: "This could be due to the presence of more large drops in LPM than PARSIVEL" – I think LPMs larger $N_w$ values are more likely owing to the much larger numbers of small drops recorded by LPM compared to the other two instruments.

48. Line 280: The last two sentences here are unclear (to which figure are the authors referring? Where is the $10^3$ number from? Which variable is affected by the imbalance the authors mention?).

49. Line 297: I think "increase in wind speed" should be "increase with wind speed" here.

50. Line 328: There are differences in the PARSIVEL and LPM values in the eyewall that are not discussed in the text.

51. Line 331: Earlier studies are mentioned but not referenced – the authors should cite them here.

**References**

Jaffrain, J., and A. Berne, 2011: Experimental quantification of the sampling uncertainty associated with measurements from PARSIVEL disdrometers. **12 (3)**, 352 – 370, doi:10.1175/2010JHM1244.1.

Raupach, T. H., M. Thurai, V. N. Bringi, and A. Berne, 2019: Reconstructing the drizzle mode of the raindrop size distribution using double-moment normalization. *J Appl Meteorol*, **58 (1)**, 145–164, doi:10.1175/jamc-d-18-0156.1.

Thurai, M., P. Gatlin, V. N. Bringi, W. Petersen, P. Kennedy, B. Notaroš, and L. Carey, 2017: Toward completing the raindrop size spectrum: Case studies involving 2D-video disdrometer, droplet spectrometer, and polarimetric radar measurements. *J Appl Meteorol*, **56 (4)**, 877–896, doi:10.1175/jamc-d-16-0304.1.

---

## Author Comment (AC1)

At the outset, the author wants to thank the reviewer for his patience in reading and suggesting improvements to the manuscript.

**Reviewer#1**

**Comment:** This manuscript describes the observations collected by three surface disdrometers (i.e., JWD, LM, and PARSIVEL) during the passage of a Tropical Cyclone. There are a few confusing sentences that need to be clarified before publishing.

*Reply: The confusing statements are rewritten with better clarity in the revised manuscript.*

**Comment:** 1. Abstract. Lines 10-12 state, "Raindrops greater than 3 mm in size are infrequent in the JWD recordings while frequent in the LPM an PARSIVEL indicating JWD underestimates the size of the raindrops than LPM and PARSIVEL due to canting of raindrops in the presence of wind." This sentence suggests the JWD underestimates raindrops greater than 3 mm diameter because the raindrops are canted in the presence of wind. This is inconsistent with conclusion #1 (lines 345-348) that states "The canting of raindrops in the presence of large horizontal winds results in more residing time in the laser beam resulting in an additional reduction in the beam intensity at the receiver. Thus, the conclusion suggests the laser disdrometers overestimate the size of the raindrops in the presence of horizontal winds." I believe the abstract needs to be corrected to match the conclusion.

*Reply: Compared to JWD, LPM and PARSIVEL disdrometers record raindrop with size greater than 3 mm. To avoid confusion, the sentence is modified as follows in the revised manuscript. LPM and PARSIVEL overestimates the raindrop size when the fall path deviates from nadir due to horizontal wind.*

**Comment:** 2. Lines 10-12 (abstract), 233-236 (body) and 345-348 (conclusion). The word "canting" only occurs in the abstract and conclusion. The body (lines 233-236) discusses why the laser disdrometers observe larger raindrops in high wind cases because the raindrops have a longer path through the laser beam. This longer path is not a raindrop canting. The three disdrometers cannot measure canting angle (the JWD measures momentum, and the two laser disdrometers only have one imaging dimension). Please clarify the manuscript and be consistent between abstract, body, and conclusions.

*Reply: I completely agree with the reviewer that canting of raindrops cannot be measured by the disdrometers used in the study. The context is to portray the deviation of raindrop fall path from nadir. Hence, in the revised manuscript the canting of raindrops is replaced with deviation of fall path from nadir.*

**Comment:** 3. Equations (1) to (6). I am confused by what processing was performed by the disdrometer and what processing was performed by the author. Please clarify in the text which processing steps described in equations (1) to (6) produced N(D) as an output from the disdrometers and which processing steps were needed to calculate N(D) off-line.

*Reply: The processing performed by the manufacturer of disdrometer indicates the converting of electrical signals into number of drops in each drop diameter interval.*

*After obtaining the number of drops information in each diameter interval, equations (1) to (6) are used to estimated N(D).*

*For better clarity, the text has been modified in the revised manuscript.*

---

## Author Comment (AC2)

At the outset, the author wants to thank the reviewer for his patience in reading and suggesting improvements to the manuscript.

**Reviewer#3**

**Comment:** The author compares at certain portions JWD with LPM and PARSIVEL and other times in vice versa. As suggested by Tokay et al. and the references therein, it could be good to keep JWD as reference and compare other two disdrometers with JWD will improve the readability of the manuscript.

*Reply: Good suggestion to improve the readability of the manuscript. In the revised manuscript, the differences are portrayed keeping JWD as reference at places wherever is required.*

**Comment:** Can the author show $Z_{DR}$ differences at different diameters at X-band? Although a reference is mentioned, but it could be good to provide the information in the manuscript.

*Reply:* $Z_{DR}$ at S-band show monotonic behaviour with raindrop diameter while at C- and X-bands show nonmonotonic behaviour. The nonmonotonic behaviour is mainly due to the resonance effect at D > 5 mm for C-band and D > 3 mm for X-band frequency radars. At resonating frequencies, the maximum deviation in $Z_{DR}$ between C-band and S-band is ~3.5 dB, between X-band and S-band is ~0.7 dB.

[Figure]

**Figure:** $Z_{DR}$ (dB) as a function of monodisperse raindrops diameter (mm) at X-band (red), C-band (green), and S-band (blue) wavelengths. For the monodisperse simulations at a drop temperature of 20 °C, the refractive index of raindrops is estimated from Ray (1972), drop axis ratio is considered from Brandes et al. (2002).

---

## Author Comment (AC3)

*At the outset, the author wants to thank the reviewer for his patience in reading and suggesting constructive improvements to the manuscript.*

Reviewer#2

**Comment:** This study shows a thorough analysis of DSD measurements from three co-located disdrometers of different types, located in Gadanki, India, during the landfall of cyclone NIVAR in November 2020. The spectra and values of key rainfall parameters are compared between the disdrometers, for different storm regions and wind speeds. The results provide a useful comparison between these three disdrometer types, and a novel aspect here is that the rainfall is not "typical" because NIVAR was a tropical cyclone.

While the comparison is thorough, it is disappointing that there is not more analysis of what the results mean in terms of the properties of the cyclone rainfall that has been sampled. The rich data source explored here would make for a useful comparison of the rainfall properties that are experienced in the different parts of the tropical cyclone - for example which parts are influenced most heavily by the drop concentration and which are influenced most heavily by drop size. Particularly concerning Figures 10, 11 and 12, the results are simply stated without physical explanations. A proper discussion of the results and their physical meanings, with references to the literature on tropical cyclone rain properties, is required. I have listed other recommendations below.

*Reply: In the revised manuscript, all the suggestions given by the reviewer are incorporated. The comparisons of different parameters over different basins are included. A detailed explanation on rain characteristics over different regions is also included in particular related to the Fig.s 10, 11 and 12.*

**Comment:** There are some grammatical errors which can affect the readability of the manuscript at times. These errors are often to do with comparisons: for example, on line 6: "high" should read "higher"; on line 7 "large" should read "larger", and so on throughout the paper. On line 61, the artefacts and errors themselves are not essential but taking them into account is essential. The use of tense in Section 3 is inconsistent. These errors are generally minor in nature and a thorough edit will fix them.

*Reply: The grammatical errors are corrected at the best in the revised manuscript.*

**Comment:** The introduction should include an introduction to what the raindrop size distribution is and its importance (to e.g. remote sensing and numerical weather prediction).

*Reply: The following text is added in introduction section of the revised manuscript.*

*"DSD is the raindrop concentration per drop size per unit volume. Spatiotemporal variations of DSD at various scales in different rain types are essential for disclosing the fundamental precipitation microphysical processes, including collision–coalescence, breakup, and evaporation (Rosenfeld and Ulbrich, 2003; Radhakrishna et al. 2020). Considering the vast application of DSD, it is one of the prime measurements required in the fields of remote sensing and numerical weather prediction."*

**Comment:** The results need to be put into more context with other studies. For example, the authors have found that different parts of the cyclone produced very different Z-R relationships. What have other authors found for cyclone Z-Rs and how do they compare to these results?

This lack of discussion extends to the other results and their physical meanings in terms of rainfall in tropical cyclones.

*Reply: In the revised manuscript, the Z-R relations are compared with the Z-R relations obtained in tropical cyclones elsewhere. The following text is added in the revised manuscript.*

*Comparing A and b values of the Z-R relations for a particular rain type in different regions provides information on precipitation microphysics. In particular. the coefficient A gives the size of raindrops, i.e., larger A for larger raindrops, and the power b provides the microphysical processes. The size or mixed controlled cases where collision-coalescence dominates the b value is greater than one and for number-controlled case (collision, coalescence, and breakup) that produces equilibrium DSD b value is ~ 1 (Atlas et al., 1999; Rosenfeld and Ulbrich, 2003). The smaller A value of LPM than PARSIVEL and JWD in all the regions indicates the overestimation of smaller drops by LPM. The retrieved b value is greater than one by all disdrometers in all the regions, suggesting the dominance of the collision-coalescence process. The Z-R relations obtained over the Atlantic basin during hurricane Anita (eyewall: $Z = 253R^{1.3}$; outer rainband $Z = 341R^{1.25}$; total $Z = 311R^{1.27}$) are given in Marks et al. (1993) and over the east pacific basin during the typhoon Lekima (Eyewall:$961.54R^{1.85}$; Inner rainband:$280.23R^{1.86}$; outer rainband:$74.25R^{1.98}$) in Bao et al. (2020), is distinctly different from the Bay of Bengal region (present study). Tropical cyclones over the Bay of Bengal and the Atlantic Ocean show an increase in A value with increasing distance from the cyclone eye while showing the opposite in the eastern Pacific basin. This confirms that the size of the raindrops and $D_m$ increases as the distance increases from the cyclone center over the Bay of Bengal and Atlantic basins, while it decreases over the eastern Pacific basin.*

**Comment:** Line 24: "Convective processes and resulting rainfall in a TC are primarily governed by the evolution of the microphysics of a TC." This statement needs a reference.

*Reply: Khain et al., (2016) references is added in the revised manuscript.*

*Khain, A., Lynn, B., and Shpund, J.P.: High resolution WRF simulations of hurricane Irene: Sensitivity to aerosols and choice of microphysical schemes, Atmospheric Research, 167, 129–145, 2016.*

**Comment:** Line 35: On underestimation of small raindrops by disdrometers, Thurai et al. (2017) also reported on this underestimation and Raupach et al. (2019) proposed a possible solution.

*Reply: The two references are added in the revised manuscript. As the disdrometers' sensitivity to smaller drops is different, and DSD varies with region a detailed assessment analysis is needs to be done before implementing the corrections and it is out of the scope of this study. Hence in this study the author is not extrapolating/correcting the data.*

**Comment:** Line 70: For the laser disdrometers was any filtering on fall velocity by drop size done, as in e.g. Jaffrain and Berne (2011)?

*Reply: The splashing and margin filler effects are removed using velocity thresholds used in Jaffrain and Berne (2011)* and *Friedrich et al. (2013) for the laser disdrometers.*

**Comment:** Line 75: i must be the diameter interval number, not the number of intervals.

*Reply: The typo error is corrected in the revised manuscript.*

**Comment:** Equation 2: What do the numbers 4600 and 1000 in this fraction

*Reply: The numbers 4600 and 1000 are constants to measure the area based on AU_{parameter} as area of LPM is device specific and varies from one instrument to another.*

**Comment:** Line 80: A reference for the LPM should be included. represent? They do not align with the given laser dimensions.

*Reply: Illingworth and Stevens (1987) reference is added in the revised manuscript.*

**Comment:** Line 93: Units should be provided for v(j).

*Reply: The unit m s^{-1} is added in the revised manuscript.*

**Comment:** Line 97: A reference for the PARSIVEL disdrometer should be provided.

*Reply: Löffler-Mang and Joss (2000) reference is added.*

**Comment:** Line 100: In Equation 5, D/2 is often used (as stated here) and yet newer PARSIVEL disdrometers automatically remove any raindrop that touches the edge of the laser area; in this case the effective sampling area should be calculated using D instead of D/2. The authors should check which is used in this case.

*Reply: As correctly pointed out by the reviewer, the LPM used in this study is not the newer version and hence D/2 is used.*

**Comment:** Equation 6: v(j) should be properly defined here to show that it refers to the jth PARSIVEL velocity class.

*Reply: To differentiate between LPM and PARSIVEL, $v_L(j)$ is used for LPM and $v_P(j)$ is used for PARSIVEL.*

**Comment:** Line 104: The locations (ie coordinates) of the disdrometers should be given, as well as their altitudes and the situation in which they are installed (e.g. open field, building roof, etc).

*Reply: The disdrometers are installed in open field. LPM and JWD are installed at 13.4608°N, 79.1733°E and PARSIVEL a 13.4565°N, 79.1758°E.*

**Comment:** Lines 106-109: How are these thresholds decided; were they based on previous studies?

*Reply: The thresholds are considered from the previous studies mentioned in Radhakrishna and Rao (2010) and references therein.*

**Comment:** Line 111: It should be noted in the paper that this 6th-DSD-moment Z is reflectivity in the Rayleigh regime, whereas the T-matrix calculations used later in the paper are in the Mie regime.

*Reply: As indicated in Fig. 10 of the revised manuscript, the T-matrix simulations used in the study show Mie regime effects only after the size of the raindrop exceeds 3 mm in diameter for X-band, and 5 mm in diameter for C-band. At S-band it shows monotonic relation till 8 mm drops indicating it is in Rayleigh regime.*

**Comment:** Line 111: $D_m$ should be labelled here as mass-weighted mean diameter.

*Reply: As Dm is already defined in lines 34-35 of the revised manuscript it will be a repetition.*

**Comment:** Equation 11: I think the $\pi^4$ in this equation should be $\pi^5$; please double check.

*Reply: $\pi^4$ is considered from Zhang et al. (2001), and Jung et al. (2008).*

*G. Zhang, J. Vivekanandan and E. Brandes, "A method for estimating rain rate and drop size distribution from polarimetric radar measurements," in* IEEE Transactions on Geoscience and Remote Sensing, *39, 4, 830-841, 2001, doi: 10.1109/36.917906.*

*Jung, Y., Zhang, G., & Xue, M. (2008). Assimilation of Simulated Polarimetric Radar Data for a Convective Storm Using the Ensemble Kalman Filter. Part I: Observation Operators for Reflectivity and Polarimetric Variables, Monthly Weather Review, 136(6), 2228-2245.*

**Comment:** Lines 140-141: The authors should reference attenuation-correction studies that use this technique here.

*Reply: Bringi et al. (1990), Jameson (1991), and Park et al. (2005) references are added in the revised manuscript.*

**Comment:** Equations 11-16: $\lambda$, K should be defined with units and the meanings of Re and Im should be written out.

*Reply: The text is modified in the revised manuscript defining $\lambda$ and K as follows.*

*"Where i stands for diameter interval, c on superscript indicates the complex conjugate, $\lambda$ is the wavelength considered, and K is the complex refractive index whose real part denotes the phase speed, and imaginary part indicates the extinction."*

**Comment:** Equations 17 and 18: $\gamma_{DP}$ and $\gamma_H$ require definitions, and these equations require better explanation.

*Reply: $\gamma_{DP}$ is the differential attenuation coefficient, $\gamma_H$ is the attenuation coefficient, and both depend on the DSD characteristics, temperature, and drop shape. As $A_{DP}$ and $A_H$ are in dB km$^{-1}$, both $\gamma DP$ and $\gamma H$ are expressed in dB per degree. This text is added in the revised manuscript.*

**Comment:** Figure 1: axis labels are missing; the Dvorak classification requires a reference on line 160; it should be stated what time interval is represented between each point that is plotted.

*Reply: The time interval is 3 h. and the following reference is added to the revised manuscript.*

*Dvorak, V. F.: Tropical cyclone intensity analysis using satellite data, vol. 11, US Department of Commerce, National Oceanic and Atmospheric Administration, 1984.*

**Comment:** Line 164: This statement about the eyewall requires a reference.

*Reply: Cecil et al., (2002) is added to the revised manuscript.*

**Comment:** Figure 2: Axes are missing labels, and the caption should state that the black solid lines show the inner/outer boundaries.

*Reply: The correction is incorporated in the revised manuscript.*

**Comment:** Line 170: What type of rain gauges were used and how close were they to the disdrometers?

*Reply: Tipping bucket rain gauge is installed near to the JWD and LPM. As the LPM observations are not there till 15 h of 25th November 2020, the author has not showed the accumulated rainfall in comparison with the rain gauge. During 25th-26th November rain gauge recorded 130 mm of rainfall whereas JWD recorded 128, and PARSIVEL 127 mm.*

**Comment:** Figure 3: It is important that the caption states the time resolution of the measurements shown here, since rain rate depends on resolution.

*Reply: The time resolution is 1-minute and is included in the revised manuscript.*

**Comment:** Line 174: The maximum $D_m$ is 2.5 mm – why do the authors discount the LPM measurement?

*Reply: 2.5 mm $D_m$ is observed only one time in the 1-minute DSD spectra so that it is discarded.*

**Comment:** Figure 4: The 1-minute resolution should also be mentioned in the caption for this figure.

*Reply: Included in the revised manuscript.*

**Comment:** Line 184: Given other studies (e.g. Thurai et al. (2017)), it is possible that the PARSIVEL has underestimated the number of small drops rather than the LPM overestimating the numbers of drops.

*Reply: One of the conclusions of this study is also illustrating the underestimation of smaller drops by PARSIVEL. However, comparison of JWD and LPM DSDs show a clear overestimation of smaller drops by LPM. Thus, the text is retained here and in the conclusions the underestimation of smaller drops by PARSIVEL than JWD is mentioned.*

**Comment:** Line 188: The authors mention corrections based on theoretical fall velocity, yet no corrections are mentioned in Section 2.

*Reply: The velocity corrections are mentioned in the revised manuscript.*

*"The splashing and margin filler effects are removed using velocity thresholds used in Jaffrain and Berne (2011), and Friedrich et al. (2013) for the laser disdrometers."*

**Comment:** Figure 5: Are these linear fits statistically significant? The authors should show significance information and discuss.

*Reply: All the fits plotted in the figures are at 95% confidence level. The information is given in the revised manuscript.*

**Comment:** Line 206: What method is used to fit the Z-R relations? If a linear relationship in log space is used it needs to be stated to distinguish the method from other methods that fit power laws specifically. The caption mentions a power-law fit but not which method was used.

*Reply: The linear fit is used in the log space and converted it into power law relation. The text is included in the revised manuscript.*

**Comment:** Line 213: It's not clear here why vertical wind speed near the surface is insignificant – I would think that vertical wind strong enough to loft 4 mm drops is easily obtained both aloft and near the surface in convective storms.

*Reply: The vertical wind at aloft can influence the fall velocity of the hydrometeors. The vertical wind greater than 2 $ms^{-1}$ sustains very minute time below 300 m altitude and persist for longer times at higher altitudes (Rogers et al., 1993). Thus, the vertical wind close to the earth surface is assumed to be minimal and its effect on drop fall velocity is neglected as raindrops of 4 mm and large require less than 12 m to attain the terminal velocity (Van Boxel et al., 1997).*

*Rogers, R. R., Ecklund, W. L., Carter, D. A., Gage, K. S., & Ethier, S. A. (1993). Research Applications of a Boundary-Layer Wind Profiler, Bulletin of the American Meteorological Society, 74(4), 567-580.*

**Comment:** Line 220: Exactly how many data points with wind over 4 m s−1 were observed? From Figure 1 it appears that this number cannot be insignificant since there are large areas where the five-minute averaged wind speed was in the 5-8 m s−1 range. Given that the event in question is a cyclone it seems reasonable that there may have been some strong winds that could skew the statistics for a $> 2$ m s$^{-1}$ wind speed category. The authors should discuss this point and justify the categories used.

*Reply: The number of 1-minute data samples observed in different wind speed intervals are depicted in below Table. AS the number of data points observed with wind speeds $> 4$ m s$^{-1}$ not sufficient to make conclusions only two intervals are considered.*

| Region | Wind speed (m s$^{-1}$) | | | |
|---|---|---|---|---|
| | 0-2 | 2-4 | 4-8 | > 8 |
| **Eyewall** | 114 | 58 | 8 | 0 |
| **Inner rainband** | 86 | 337 | 616 | 41 |
| **Outer rainband** | 964 | 650 | 6 | 0 |

**Comment:** Line 222: It would be helpful to briefly explain the DSD classification used here.

*Reply: DSDs are not classified into different categories. Based on the cyclone position the DSD data are grouped to eyewall, inner and out rainbands.*

**Comment:** Line 226: Rainfall variability is high enough that even "co-located" instruments metres apart sample different rainfall properties, so not all differences can be put down to instrument error or measurement principle.

*Reply: One to one comparison of 1-minute DSDs shows variability between co-located instruments. However, comparing the DSDs at event level will minimize these variabilities.*

**Comment:** Figure 6: It is possible that extreme values skew the mean DSDs shown here. The authors should test whether the median DSDs are very different - if they are, then showing the median DSDs may be more representative of the "characteristic" DSD.

*Reply: Non-Gaussian distributions, the median is better representative to characterize the distributions. However, the mean DSDs shown in Fig. 6 are the same with median. Hence the mean DSDs are shown in Fig. 6.*

**Comment:** Line 230: Again, I wonder whether what the authors call an "overestimation" by the LPM is actually an underestimation by PARSIVEL and JWD?

*Reply: As mentioned in Tokay et al. (2008) and another reviewer, JWD is considered as standard reference to compare the DSDs in this study. Thus, comparing the JWD, LPM overestimates the smaller raindrops. This is also proven by comparing the particle measuring system observations elsewhere.*

**Comment:** Line 234: The different properties of JWD underestimation in different storm regions makes me wonder whether the physical set-up of the instrument could play a role - i.e. if there is a nearby building wind direction could make a difference.

*Reply: The JWD is located in the open field and no building is present near to the location.*

**Comment:** Line 233: The JWD also records more large drops than the other instruments in the inner rainband for low rain rates.

*Reply: As seen from Fig. 6, the overestimation of medium sized raindrops is obvious at wind speeds $> 2$ m s$^{-1}$ in the inner rainband. At smaller wind speeds the it is not clear. Thus, in the text it is not mentioned.*

**Comment:** Lines 240-245: It is unclear here how the authors are judging whether a difference in slope between the different plots is significant or not, and this should be stated. For example, the difference between JWD lines in the eyewall is only slightly larger than differences in the other storm regions. The authors should also use language that acknowledges the uncertainty – ie use "little difference" instead of "no difference" when the differences are not significant, and "similar" instead of "same" distribution, since there are still differences present.

*Reply: The text is modified in the revised manuscript as follows.*

*"The effect of wind speed is not uniform for all the disdrometers in different regions of a TC. For a given R, JWD shows an increase in $D_m$ with wind speed in the eyewall region, while small variation in $D_m$ with the wind in the inner and outer rainbands. PARSIVEL data show an increase in $D_m$ with the wind in the eyewall, a decrease in $D_m$ with the wind in the inner rainband, and minor variations in the outer rainband. LPM shows an increase in $D_m$ with the wind in the eyewall and inner rainband and small variations in the outer rainband. The observed differences in the $D_m$-R relations under the same environmental conditions indicate that the DSD spectra recorded by three disdrometers are different."*

**Comment:** Line 252: Do the authors mean at wind speed $> 2$ m s$^{-1}$ instead of R $> 2$ mm h$^{-1}$?

*Reply: The text is modified in the revised manuscript for better clarity,*

**Comment:** Lines 255-265: The repetitive nature of the results shown here makes this section difficult to read and it is unclear whether the authors are referring to Figure 7 or Figure 6 or to both figures. The discussion could be made more concise.

*Reply: The text is modified as follows. "At medium and large raindrops, the raindrop concentration observed by PARSIVEL and LPM is similar and lower than the JWD. Thus, at all wind speeds with R $< 5$ mm h$^{-1}$, the $D_m$ values are small for PARSIVEL and large for JWD in the inner rainband. At higher rain intensities, LPM overestimates the small raindrop concentration (by two orders of magnitude), while both LPM and PARSIVEL underestimates*

*the medium-sized and overestimates the large-sized raindrops than JWD. The imbalance between the small, medium, and large raindrops results in large $D_m$ values for JWD at all wind speeds, while for LPM small $D_m$ values at wind speed less than 2 m s$^{-1}$, and large $D_m$ values at higher wind speeds than for PARSIVEL in the inner rainband. Although LPM and PARSIVEL show nearly the same distribution at the medium and large raindrops in the outer rainband, LPM overestimates the small raindrops, resulting in marginally smaller $D_m$ than PARSIVEL at all R and wind. Compared to JWD, LPM and PARSIVEL records a high concentration of small and large raindrops and a low concentration of medium-sized raindrops at all R and wind, which imbalance the DSD spectrum to produce marginally small $D_m$ than JWD in the outer rainband."*

**Comment:** Lines 259-261: This sentence reports on all wind speeds but also wind speeds less than 2 m s$^{-1}$ which does not make sense.

*Reply: At all wind speed for JWD, while for different for different wind speeds for LPM when compared to PARSIVEL. The sentence is reframed for better clarity.*

**Comment:** Line 263: The claim that LPM shows smaller $D_m$ than PARSIVEL in the outer rain band is not supported by Figure 7.

*Reply: As indicated in Figs. 6 & 7, the underestimation of smaller and larger drops by PARSIVEL than LPM results in smaller $D_m$ values of PARSIVEL than LPM. The text is modified accordingly in the revised manuscript.*

**Comment:** Line 267: 10 log$_{10}$N$_w$ should be written 10 log10 Nw.

*Reply: The typo error is corrected.*

**Comment:** Line 269: The statement that "In general, Nw increases with increasing R" is not supported by Figure 8. Do the authors mean that they expect Nw to increase with R, given previous work?

*Reply: Yes, as per Testud et al. (2001), Nw should increase with R. However, it is not always true. As shown in Ma et al. (2019) the imbalance between the small, medium and large diameters alters the $N_w$ trend with R.*

**Comment:** Line 272: "$N_w$ is smaller in the eyewall while larger in the inner and outer rainbands than at lower wind speeds" - this statement contradicts Figure 8 which shows that $N_w$ is generally larger in the eyewall than in other storm regions. The authors meaning here is unclear.

*Reply: To avoid the confusion the sentence modified in the revised manuscript.*

**Comment:** Figure 8: The statistical significance of the linear fits should be discussed. It may well be that some of these fits are not significant enough to show an increase or decrease of $N_w$ with R, since by eye the slopes often look close to zero.

*Reply: JWD shows an increase in $N_w$ with R in the inner and outer rainbands while a decrease in the eyewall at all wind speeds. The decrease in $N_w$ with R is small at lower wind speed and considerable at higher wind speeds. PARSIVEL measurements indicate an increase in $N_w$ with R in the eyewall and outer rainbands while a decrease in the inner rainband. The change in $N_w$ with R is considerable at all wind speed in all the regions of a TC except at low wind speeds in the outer rainband and high wind speeds in the inner rainband. LPM data show an increase*

*in $N_w$ with R in the outer rainband and a decrease in the inner rainband while increasing at low wind speeds and decreasing at high wind speeds in the eyewall. A sizable change in $N_w$ with R is observed in the inner rainband and at high winds in the outer rainband and small in the eyewall and low windspeeds in the outer rainband.*

**Comment:** Line 277: "This could be due to the presence of more large drops in LPM than PARSIVEL" – I think LPMs larger $N_w$ values are more likely owing to the much larger numbers of small drops recorded by LPM compared to the other two instruments.

*Reply: $N_w$ values are larger for PARSIVEL than LPM. As shown in Ma et al. (2019) this could be due to the presence of less large drops in PARSIVEL than LPM.*

**Comment:** Line 280: The last two sentences here are unclear (to which figure are the authors referring? Where is the $10^3$ number from? Which variable is affected by the imbalance the authors mention?).

*Reply: The text is modified for better clarity in the revised manuscript.*

**Comment:** Line 297: I think "increase in wind speed" should be "increase with wind speed" here.

*Reply: The correction is incorporated in the revised manuscript.*

**Comment:** Line 328: There are differences in the PARSIVEL and LPM values in the eyewall that are not discussed in the text.

*Reply: The differences between PARSIVEL and LPM in the eyewall are discussed in the revised manuscript.*

**Comment:** Line 331: Earlier studies are mentioned but not referenced – the authors should cite them here.

*Reply: The references are added to the revised manuscript.*

**Comment:** References

Jaffrain, J., and A. Berne, 2011: Experimental quantification of the sampling uncertainty associated with measurements from PARSIVEL disdrometers. 12 (3), 352 – 370, doi:10.1175/2010JHM1244.1.

Raupach, T. H., M. Thurai, V. N. Bringi, and A. Berne, 2019: Reconstructing the drizzle mode of the raindrop size distribution using double-moment normalization. J Appl Meteorol, 58 (1), 145–164, doi:10.1175/jamc-d-18-0156.1.

Thurai, M., P. Gatlin, V. N. Bringi, W. Petersen, P. Kennedy, B. Notaroˇs, and L. Carey, 2017: Toward completing the raindrop size spectrum: Case studies involving 2D-video disdrometer, droplet spectrometer, and polarimetric radar measurements. J Appl Meteorol, 56 (4), 877–896, doi:10.1175/jamc-d-16-0304.1.

*Reply: These references are cited in the revised manuscript.*

---

## Author Response (AR2)

*At the outset, the author wants to thank the reviewer for his patience in reading and suggesting improvements to the manuscript.*

**Reviewer#2**

**Comment:** Line 65: The errors and artefacts are not essential, understanding them is. Please rephrase.

*Reply: The text is modified as follows.*

*"The artefacts and instrument errors associated with various kinds of disdrometers mentioned above need to be quantified as they propagate to the retrievals of radar geophysical parameters (Adirosi et al., 2018) and, in turn, in surface rainfall from weather radars (both polarimetric and non-polarimetric)."*

**Comment:** Line 76: Please include a reference for these facts about JWD.

*Reply: The reference Joss and Waldvogel (1967), is added in the revised manuscript.*

**Comment:** Line 118: The authors state ``The thresholds are considered from the previous studies mentioned in Radhakrishna and Rao (2010) and references therein" yet the reference is not included in the manuscript. Please include it here to justify the threshold choices.

*Reply: The reference is added in the revised manuscript.*

**Comment:** Line 125: As I mentioned, the authors need to state that this equation for reflectivity is only for the Rayleigh regime.

*Reply: The text is included in the revised manuscript.*

*"The estimated N(D) is used to calculate rain rate (R), reflectivity (Z) assuming Rayleigh approximation, $D_m$, and normalized intercept parameter ($N_w$) using the following relations."*

**Comment:** Line 149: The wavelength requires a unit in this description.

*Reply: the units are added to the revised manuscript.*

**Comment:** Figure 1: The figure requires axis labels.

*Reply: The axis labels are added to Fig. 1.*

**Comment:** Figure 8: The significance of the fitted lines is not discussed -- all fits in the paper should have their statistical significance indicated.

*Reply: The significance levels are mentioned in the figure captions wherever (Figs. 7, 8, and 9) required.*

*At the outset, the Author profoundly thanks Dr. Zamin A. Kanji, Associate editor, AMT for his positive and timely responses and handling of this manuscript reviews.*

**Associate Editor**

**Comments to the author**: As seen from the positive reviewer reports, they are satisfied with the revisions you have made to the manuscript. Reviewer 2 has recommended minor revisions. Please address those concerns before we can proceed further with publication. These concerns are small and should not take much time. In case you disagree with any of the comments, please submit a response explaining why. This will be important to move forward with accepting the manuscript for final publication in AMT.

***Reply:*** *All the comments/suggestions of Reviewer#2 are implemented in the revised manuscript.*